**Resource**

# Characterization of subclonal variants in HG002 Genome in a Bottle reference material as a resource for benchmarking variant callers

## Graphical abstract

## Authors

Camille A. Daniels, Adetola A. Abdulkadir, Megan H. Cleveland, ...,
Semyon Kruglyak, Justin M. Zook,
Nathan D. Olson

## Correspondence

justin.zook@nist.gov (J.M.Z.),
nolson@nist.gov (N.D.O.)

## In brief

Daniels et al. extended the Genome in a Bottle HG002 characterization to include subclonal variants, enabling the development of a new benchmark set for evaluating somatic and mosaic variant calling methods.

## Highlights

- Expanded Genome in a Bottle HG002 RM characterization to include subclonal SNVs

- Subclonal variants were identified using a trio-based method

- Benchmark set for evaluating somatic and mosaic variant detection methods

- The benchmark set includes 85 subclonal SNVs and covers 2.45 Gbp of the genome

 Daniels et al., 2026, Cell Genomics 6, 101104
April 8, 2026 Published by Elsevier Inc.

# Cell Genomics

## Resource

# Characterization of subclonal variants in HG002 Genome in a Bottle reference material as a resource for benchmarking variant callers

Camille A. Daniels,[1] Adetola A. Abdulkadir,[1] Megan H. Cleveland,[2] Jennifer H. McDaniel,[2] David Jáspez,[3] Luis Alberto Rubio-Rodríguez,[3] Adrián Muñoz-Barrera,[3] José Miguel Lorenzo-Salazar,[3] Carlos Flores,[3,4,5,6] Byunggil Yoo,[7] Sayed Mohammad Ebrahim Sahraeian,[8] Yina Wang,[9] Massimiliano Rossi,[9] Arun Visvanath,[9] Lisa Murray,[9] Wei-Ting Chen,[9] Severine Catreux,[9] James Han,[9] Rami Mehio,[9] Gavin Parnaby,[9] Andrew Carroll,[10] Pi-Chuan Chang,[10] Kishwar Shafin,[10] Daniel Cook,[10] Alexey Kolesnikov,[10] Lucas Brambrink,[10] Mohammed Faizal Eeman Mootor,[11] Yash Patel,[11] Takafumi N. Yamaguchi,[11] Paul C. Boutros,[11] Karolina Sienkiewicz,[12,13] Jonathan Foox,[12] Christopher E. Mason,[12,13,14] Bryan R. Lajoie,[15] Carlos A. Ruiz-Perez,[15] Semyon Kruglyak,[15] Justin M. Zook,[2,16,17,*] and Nathan D. Olson[2,16,17,18,*]

[1]Medical Device Innovation Consortium (MDIC), 1655 N Ft. Myer Drive, Suite 250, Arlington, VA 22209, USA
[2]Material Measurement Laboratory, National Institute of Standards and Technology (NIST), 100 Bureau Dr, MS8312, Gaithersburg, MD 20899, USA
[3]Genomics Division, Instituto Tecnológico y de Energías Renovables (ITER), Santa Cruz de Tenerife, Spain
[4]Research Unit, Hospital Universitario Nuestra Señora de Candelaria, Instituto de Investigación Sanitaria de Canarias, Santa Cruz de Tenerife, Spain
[5]CIBER de Enfermedades Respiratorias (CIBERES), Instituto de Salud Carlos III, Madrid, Spain
[6]Facultad de Ciencias de la Salud, Universidad Fernando de Pessoa Canarias, Las Palmas de Gran Canaria, Spain
[7]Genomic Medicine Center, Children's Mercy Kansas City, Kansas City, MO, USA
[8]Roche Sequencing Solutions, Santa Clara, CA 95050, USA
[9]Illumina, Inc., San Diego, CA, USA
[10]Google, Inc., Mountain View, CA, USA
[11]Department of Human Genetics, University of California, Los Angeles, Los Angeles, CA, USA
[12]Department of Physiology and Biophysics, Weill Cornell Medicine, New York, NY, USA
[13]The HRH Prince Alwaleed Bin Talal Bin Abdulaziz Alsaud Institute for Computational Biomedicine, Weill Cornell Medicine, New York, NY, USA
[14]The WorldQuant Initiative for Quantitative Prediction, Weill Cornell Medicine, New York, NY, USA
[15]Element Biosciences, San Diego, CA, USA
[16]Senior author
[17]These authors contributed equally
[18]Lead contact
*Correspondence: justin.zook@nist.gov (J.M.Z.), nolson@nist.gov (N.D.O.)

## SUMMARY

We developed a benchmark set of subclonal variants in the Genome in a Bottle (GIAB) Consortium HG002 reference material (RM) DNA for evaluating lower-frequency variant callsets. We used a somatic variant caller with high-coverage (300×) whole-genome sequencing data from the GIAB Ashkenazi Jewish trio to identify potential subclonal variants in the HG002 RM DNA. Using orthogonal sequencing data and manual curation, we defined a benchmark set with 85 high-confidence subclonal single-nucleotide variants (SNVs) (allele frequency [AF] > 5%) and a benchmark region covering 2.45 Gbp of the autosomes. External validation supported that it can be used to reliably identify both false negatives and false positives for a variety of sequencing technologies and variant callers. By adding our characterization of mosaic SNVs in this widely used cell line, we have expanded the scope of bioinformatic and sequencing applications for which the HG002 GIAB RM can be used to include benchmarking subclonal SNVs.

## INTRODUCTION

Germline variant calling in human genome studies typically targets heterozygous and homozygous variants occurring at variant allele frequencies (VAFs) of 50% or 100%, respectively. However, variants can occur at lower frequencies if they are only present in a subset of cells due to somatic mosaicism, making them harder to detect and requiring different variant calling methods to identify and characterize them. Somatic mutations occur within a genome after conception, are typically not

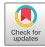

inherited, and are only present in a subset of cells. While many of these mutations are non-pathogenic, others can cause unrestricted cell growth and lead to cancer or play roles in the development of neurodegenerative, monogenic, and complex diseases.[1,2] An initiative by the National Institutes of Health (NIH) Common Fund called Somatic Mosaicism across Human Tissues (SMaHT; https://smaht.org/) has been instituted to establish a repository of mosaic variants from various healthy tissue types and address the lack of resources to study somatic mosaicism. This effort has recognized a need for benchmarks to evaluate and validate low-frequency variant calling methods. In this work, we use the term "mosaic variants" to mean variants present in some but not all cells in a large batch of DNA from the deeply characterized Genome in a Bottle (GIAB) normal lymphoblastoid cell line HG002.

Previous GIAB benchmarks using whole-genome sequencing (WGS) have focused on characterizing germline small[3–8] and structural[8–10] variants, generally ignoring variants with a <30% VAF. These benchmark sets include high-confidence variant calls and regions. The benchmark variants are confident homozygous and heterozygous variants in a sample relative to a reference genome (VCF file), and the benchmark regions (BED file) are genomic regions that are confidently identified as a homozygous reference or a benchmark variant. The benchmark variants enable users to identify true positives and false negatives in their query callset, and benchmark regions enable the identification of false positive variants. The reliable identification of errors (RIDE) principle is used to determine if a GIAB benchmark set is fit for purpose, specifically the identification of false positives and false negatives across a variety of high-quality methods.[11] In addition to the GIAB benchmark sets, the GIAB Consortium has worked with the Global Alliance for Genomics and Health (GA4GH) to define best practices for benchmarking small variants.[12] While the GIAB benchmark sets and benchmarking methods have been used to evaluate small and structural variant calling methods, as well as training machine-learning- and deep-learning-based variant calling methods, low-frequency or mosaic variants in GIAB reference materials (RMs) have not been previously characterized.

Characterization of mosaic variants in GIAB RMs would allow researchers to use the GIAB RMs and genome sequencing data generated from the material to validate mosaic and somatic variant calling methods, as well as other uses, such as negative controls when evaluating methods for detecting off-target genome edits. To ensure that next-generation sequencing (NGS) protocols and bioinformatic pipelines can accurately and reliably detect low-frequency mutations, well-characterized reference samples are needed. Previous efforts developing benchmarks for low-frequency variants have used a variety of strategies. For example, data from four historical cell lines were mixed in a variety of ratios to mimic mosaic variants at different frequencies.[13,14] However, because these are germline variants, some mosaic and somatic variant callers will filter them, and there is a lack of clarity in accurately identifying true negatives. Therefore, cancer-focused benchmarks have taken other approaches, such as injecting synthetic somatic variants into real data,[15] simulating tumor subclonality,[16] creating synthetic DNA with somatic mutations spiked into a normal background

sample,[17] engineering normal samples to contain somatic variants,[18] comparing paired tumor and normal cell lines derived from a single individual,[19,20] and creating mixtures of cell lines from different individuals.[21]

Here, we complement these previous efforts by leveraging publicly available homogenous batches of DNA RMs linked to explicit consent for public genome data sharing. By characterizing the baseline mosaic variants in this cell line, it can be used more robustly as a negative control or background in many of the other benchmarking approaches (e.g., when modifying reads to contain mutations, adding spike-in DNA with mutations, or mixing with other samples). Specifically, we present an initial mosaic benchmark set for the GIAB HG002 RM from a broadly consented individual from the Personal Genome Project.[22] The GIAB RM used for this study originates from a large homogenous batch of DNA isolated from the HG002 cell line.[4] Mosaic variants in this RM DNA may be from somatic mosaicism in the individual's B cells or from mutations that have arisen during the cell line generation and culturing process. To generate this benchmark, we used a trio-based approach (Figure 1). We first identified potential mosaic variants using the $300\times$ coverage Illumina sequencing data and the Strelka2 tumor/normal somatic variant caller with the son (HG002) as the tumor sample and the parents (HG003 + HG004) as the normal samples. High-coverage orthogonal sequencing data for the NIST HG002 RM DNA were used to validate the low-frequency variant calls identified by Strelka2. While there is some preliminary evidence of possible mosaics occurring above a 30% VAF in human genome data, this study focused on somatic mosaic variants between 5% and 30% VAFs.

## RESULTS

### Mosaic benchmark set generation

The HG002 mosaic benchmark set includes 85 validated and manually curated SNVs and benchmark regions covering 2.45 Gbp (Figure 2). To arrive at this benchmark, first, potential mosaic variants in the HG002 NIST RM DNA were identified using the $300\times$ Illumina Ashkenazi Jewish (AJ) trio dataset and the Strelka2 somatic variant caller, with HG002 (son) data as the tumor sample and HG003 + HG004 (parents) data as the normal samples. The Strelka2 callset contained $\approx$1.27 million passing (PASS) and filtered SNVs and insertions or deletions (indels), with 425,679 potential mosaic variants after excluding variants in the AJ trio v.4.2.1 GIAB small variant benchmark. Exclusion of the AJ trio v.4.2.1 complex and structural variants in HG002 further reduced the number of potential somatic variants to 366,728 (see Figure S1). While only 1,916 SNVs and 21 indels passed the Strelka2 filter, we kept the filtered variants for downstream analysis (see Table 1) to reduce the probability of missing variants based on our in silico mixture experiments demonstrating 99.4% recall for SNVs at a 5% VAF (see Figure S2).

To define our mosaic benchmark variants and regions, we evaluated support for the 366,728 potential mosaic variants across multiple short- and long-read technologies with at least $100\times$ coverage per technology. We only used technologies with at least $100\times$ coverage to ensure sufficient power to detect variants with a >5% VAF, and all data were from the NIST RM

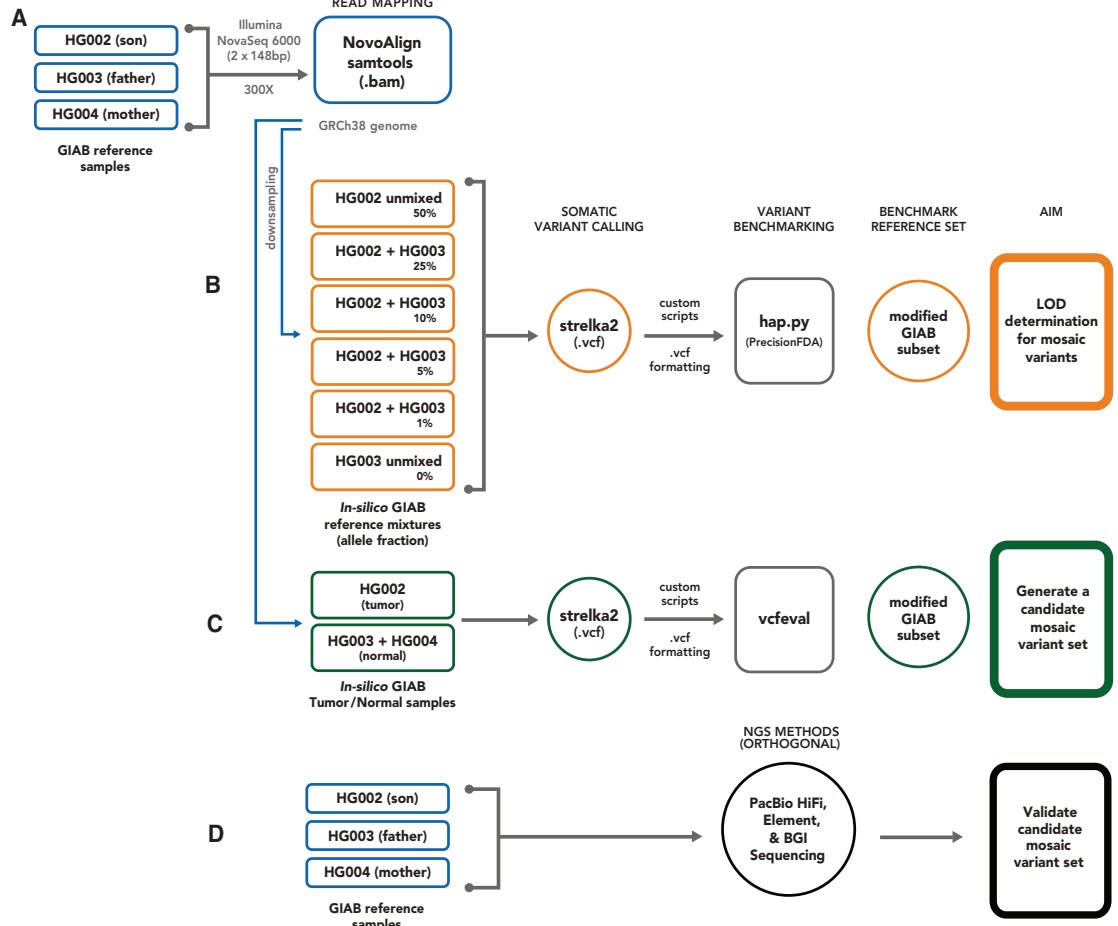

**Figure 1. Trio-based methodology using high-coverage Illumina data, Strelka2 somatic caller, and orthogonal next-generation sequencing datasets for candidate mosaic variant detection and validation in HG002**

(A) Ashkenazi Jewish trio (NIST RM: HG002, HG003, and HG004) sequencing and reference mapping (GRCh38) were initially performed by Zook et al.[4]

(B) *In silico* sample mixtures were created using HG002 and HG003, treating HG003 as normal and the mixtures as tumor, to determine the limit of detection (LOD) for variant allele frequencies. Strelka2 somatic calling and benchmarking with hap.py were conducted using the GIAB mixtures to estimate a LOD.

(C) To identify potential mosaic and *de novo* variants, a tumor-normal Strelka2 somatic run, with HG002 (son) as tumor and HG003+HG004 (combined parents) as normal, was performed.

(D) The Strelka2 callset was benchmarked against the GIAB v.4.2.1 small variant benchmark with vcfeval to create a candidate variant set, and three orthogonal high-coverage short- and long-read sequencing technologies were used for validation.

DNA to avoid batch effects. We created a database with these potential mosaics that included Strelka2 VCF annotations, read support metrics from multiple sequencing technologies, and genomic context (data and code availability). Using an initial set of heuristics (see Figure S3), we filtered this database and identified 135 variants for manual curation. More than 98% of the candidates were removed due to very low VAFs (less than 3% with 99% confidence). After manual curation, an additional 50 variants were excluded due to a lack of long-read support, high-coverage short-read support, or low VAFs reported by Strelka2 (Figure 2; Table S2). For variants excluded due to a lack of long-read support, most were detected in one or both parents (HG003 and/or HG004) and were in segmental duplication regions associated with mapping errors and copy-number variation. To create the HG002 mosaic benchmark v.1.0 BED

file, we excluded genomic regions with tandem repeats and homopolymers, regions containing variants that could not be confidently determined to be >5% or <2% VAFs (see Figure S3, yellow squares), and regions <50 bp, since small benchmark regions can cause problems with benchmarking complex variants.

**Benchmark set characterization**

The HG002 mosaic benchmark v.1.0 contains 85 SNVs with VAFs between 5% and 30% (Figure 3A, 99% one-sided confidence interval $\geq$ 0.05, calculated based on the combined read support across multiple orthogonal sequencing technologies; Figures 2 and S4; data and code availability). The mosaic benchmark included 2.45 Gbp (89.5% of the GRCh38 non-gapped assembled bases in the autosomes) with 891,240 regions,

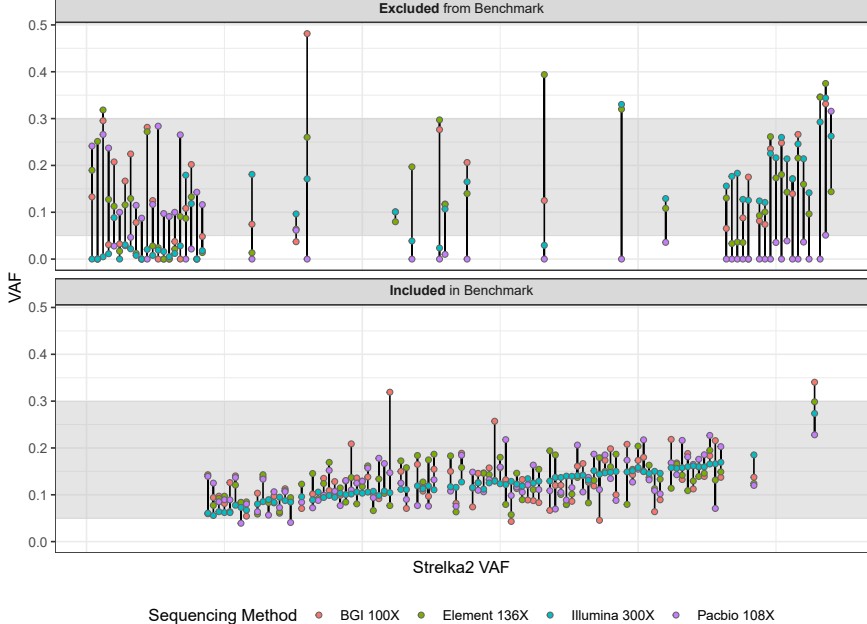

**Figure 2. Variant allele frequency of manually curated potential mosaic variants (135) for orthogonal sequencing methods**

Variant allele frequency (VAF) of multiple sequencing methods depicted as points on vertical lines and ordered by increasing Strelka2 VAF (x axis). The shaded area represents VAFs (5%–30%), the targeted VAF for inclusion in the benchmark. Curated variants were excluded from the benchmark set (top) for multiple reasons (see the mosaic benchmark set generation section of the methods and see Figure S3), often due to VAFs below 5% for one or more sequencing technologies (e.g., lack of support from long reads). Nearly all benchmark variants (bottom) had VAFs between 5% and 30% for all four sequencing methods.

ranging from 50 bp to 76.6 kbp (data and code availability). All the candidate indels identified during the benchmark set generation process were excluded by our heuristics, and therefore, the benchmark set only includes SNVs. The benchmark set included a few variants in challenging genomic regions, with two of the 85 variants near homopolymers and two in low-mappability regions (Figure 3B). While no mosaic benchmark variants were observed in GRCh38 coding regions, 13 of 85 (~15%) variants are in medically relevant genes (MRGs),[8] and benchmark regions included >90% of the bases in these genes (see Table S3). The HG002 mosaic benchmark regions include a total of 417,752,807 bases in MRGs and >90% of bases in 3,871 MRGs.

The mosaic benchmark set provides a characterization and VAFs for variants in the NIST RM 8391 DNA, but these can differ in cells available from other batches of HG002. For example, we identified differing VAF profiles for the benchmark mosaic variants between the large batch of DNA distributed as NIST RM 8391 and various unknown batches of non-RM DNA from Coriell (GM24385/NA24385). For both Element and PacBio Revio, the mosaic benchmark VAFs were significantly higher than the datasets generated using non-RM DNA; these differences are likely to result from changes in the cell line rather than random sampling (Figure 4). The larger difference for PacBio Revio suggests additional clonal changes prior to its sequencing. While variants differed in VAFs between batches, they were generally present in all batches. For accurate benchmarking, users will want to use variants called using sequencing data generated from the NIST RM, as other batches may have different VAF profiles or sets of low-frequency variants.

**Mosaic benchmark set external validation**

To validate that the mosaic benchmark set can be used to reliably identify errors (RIDE principle), we compared the draft benchmark set to variant callsets submitted by external evalua-

tors and manually curated the discrepancies, ensuring the discrepancies are errors in the comparison callset and not the benchmark (Table 2). Eight somatic variant calling groups submitted callsets for use in validating the draft mosaic benchmark. Most groups used the same GIAB HG002 Illumina 300× data used to generate the mosaic v.1.0 benchmark, one group provided an in-house HG002 40× callset, and others produced additional short-read (Element 70× and 100× and PacBio Onso 35×) and long-read (PacBio Revio 130×) HG002 callsets (see Table S4). The evaluations focused on curating differences between external callsets and a draft version of the mosaic benchmark generated from both HG002 RM (NIST RM 8391) and non-RM (Coriell, NA24385) data. Unlike the final benchmark, the draft benchmark included some non-RM data because RM data from some technologies were not yet available.

During the evaluation, we found that the draft mosaic benchmark variants were reliable, but some variants were incorrectly filtered by our initial heuristics. For example, we had initially ignored candidate mosaic variants with lower-than-normal coverage, as low-coverage regions were generally associated with alignment errors around larger germline variants. However, some true mosaic variants were ignored but kept in the benchmark regions, so we modified the heuristics for v.1.0 to only ignore variants below the 0.5% quantile in coverage for all datasets combined or HiFi only. Additionally, the draft benchmark missed some variants substantially different in VAFs between RM and non-RM DNA, so we only used RM DNA data for the v.1.0 benchmark.

We observed a single case of a true mosaic SNV missed by our trio-based approach, identified by one method using only HG002 data (chr10:106867519). Interestingly, this variant appears to be a heterozygous germline variant in the father (HG003), but HG002 inherited the reference allele from HG003 and HG004, so it appears to be a true mosaic variant in HG002 that matches the father's variant but occurred independently (see Figure S5), which has been seen previously.[24] This variant and the adjacent 50 bp on either side were not included in the benchmark VCF or BED files, so the parents can be used as

**Table 1. GIAB AJ trio (HG002: son, HG003: father, and HG004: mother) and orthogonal datasets for HG002 mosaic benchmark generation**

| Library | NIST ID | Coverage | SRA accession no. | Publication |
|---|---|---|---|---|
| Illumina 2 × 148 bp | HG002 | 300× | SRA: SRX847862–SRX848005 | Zook et al.[4] |
| Illumina 2 × 148 bp | HG003 | 300× | SRA: SRX848006–SRX848173 | Zook et al.[4] |
| Illumina 2 × 148 bp | HG004 | 300× | SRA: SRX848174–SRX848317 | Zook et al.[4] |
| BGI 2 × 150 bp | HG002 | 100× | SRA: SRX22242218 | this study |
| Element 2 × 150 bp, standard | HG002 | 81× | SRA: SRR30945810 | this study |
| Element 2 × 150 bp, long insert | HG002 | 55× | SRA: SRR30945809 | this study |
| PacBio HiFi Sequel 10 kb | HG002 | 30× | SRA: SRX5327410 | Wenger et al.[23] |
| PacBio HiFi Sequel 15 kb | HG002 | 28× | SRA: SRX6908796, SRX6908797, SRX6908798 | unpublished |
| PacBio HiFi Revio 20 kb | HG002 | 48× | N/A | unpublished[a] |

Sequencing data were generated using NIST reference material (RM) DNA (NIST RM 8391 for HG002 and NIST RM 8392 trio). See key resources table for non-reference material (Coriell, NA24385) datasets.
[a]Data are publicly available on the PacBio website (https://www.pacb.com/connect/datasets/) as well as GIAB FTP site (https://ftp-trace.ncbi.nlm.nih.gov/ReferenceSamples/giab/data/AshkenazimTrio/HG002_NA24385_son/PacBio_HiFi-Revio_20231031).

normal when evaluating tumor/normal somatic variant callers. It is possible there may be additional mosaic variants like this missed by the benchmark if they coincide with parental germline variants, but this was the only one identified during the external evaluation.

The external evaluation results informed refinement of the draft benchmark into v.1.0, which considered only NIST RM 8391 data, by adding 15 mosaic SNVs no longer filtered by our heuristics, for a total of 85 SNVs in the v1.0 HG002 mosaic

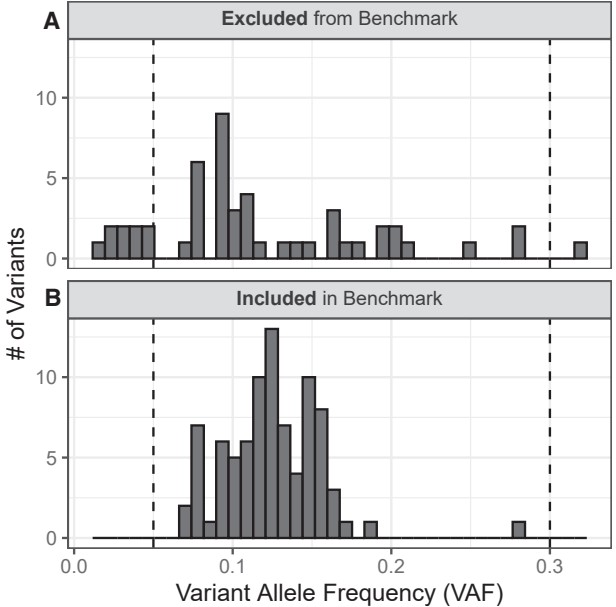

**Figure 3. SNV variant allele frequencies (VAFs) for manually curated mosaic benchmark variants**
Based on manual curation observations, variants were excluded from (A) and included in (B) the benchmark. Values represent VAFs combined across all orthogonal technologies (BGI, Element, Illumina, PacBio Revio, and Sequel). The dashed vertical lines represent the targeted VAF range (5%–30%).

benchmark set. Using hap.py to compare the external callsets to the v.1.0 benchmark, most mosaic benchmark variants (≥87%) were present in short- and long-read callsets by the six groups that used high-coverage datasets. A total of 16 variants in the benchmark set were missed by at least one external callset (using >100× coverage data as input), and 11 of the 16 were missed by only one short-read callset. External and internal groups manually confirmed that all 16 variants had VAFs within our level of detection (5%–30% VAFs) and were true mosaic variants.

Most variants in the external validation callsets not present in the v.1.0 benchmark had VAFs below 5%, our established limit of detection (LOD), and therefore could be false positives or true low VAF variants. We further curated 163 SNVs with VAFs between 5% and 30% in external callsets but not in the draft benchmark and found that all but two were likely mapping errors, due to segmental duplications or copy-number variants in HG002, systematic sequencing errors, local alignment errors around germline insertions, or batch effects (for callsets from non-RM sequencing data) (Table 2; Figure 5; see also Table S5). One SNV (chr6:150458314) is likely a true mosaic variant near a 4 bp insertion on the other haplotype that was missed by our mosaic benchmark but was present in callsets from all eight external validator groups. The other remaining variant (chr1:242208421) is likely a true mosaic variant, though it likely has a <5% VAF in the RM DNA. These two variants, along with 50 bp flanking sequences, were removed from the mosaic v.1.0 bed to generate HG002 mosaic benchmark v.1.1.

## DISCUSSION

GIAB benchmark sets have focused on germline variants with VAFs 50% or 100%, which are higher than typical mosaic or somatic variants. To address the need for a benchmark for variants with lower VAFs, which occur in only a fraction of the cells, we developed the first GIAB mosaic SNV benchmark set for the highly characterized HG002 genome using data from the GIAB Ashkenazi Jewish (AJ) trio (son and parents). We substituted a combined

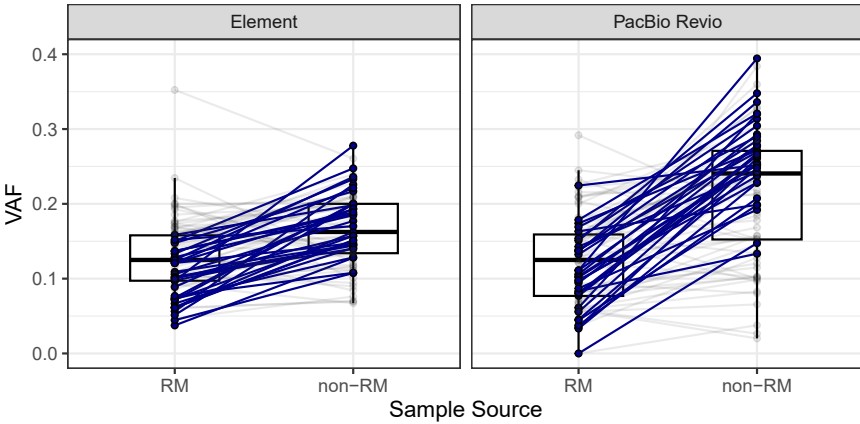

**Figure 4. Mosaic variants change VAFs between batches of DNA**
HG002 mosaic benchmark variant allele frequencies (VAFs) for NIST reference material (RM) 8391 and different batches of non-RM DNA (Coriell, NA24385) for two orthogonal technologies (Element and PacBio Revio). Significantly higher VAFs (indicated by blue lines) were observed in direct VAF comparisons between materials compared to the GIAB reference material. Coverage: Element RM, 81× and non-RM, 100×; PacBio Revio RM, 48× and non-RM, 120×.

parental BAM for the typical matched normal sample (Figure 1C) and followed best practices for somatic variant calling[25] and developing benchmarks[11] by filtering out normal variants and artifacts, performing manual curation to confirm the candidate mosaic set, and comparing the benchmark with orthogonal methods.

We identified 85 benchmark SNVs in the HG002 RM DNA with VAFs between 5% and 30% using high-coverage Illumina and orthogonal short- and long-read methods (BGI, Element, and PacBio HiFi) (Figure 2; Table S1). External validation confirmed that the benchmark set can be used to reliably identify errors in somatic and mosaic variant callsets. Variants were included in the HG002 mosaic benchmark v.1.0 for several reasons: (1) they passed decision tree heuristics that required sufficient confidence, where the variants had a >5% VAF to be included or a <2% VAF to be ignored, (2) there was support in long reads for difficult-to-map regions, (3) confirmation by manual curation was possible, and (4) they represented typical coverage for the region.

While variants with <5% VAFs were not retained in this mosaic benchmark due to the LOD of our discovery approach, further investigation using library preparation that corrects DNA damage due to extraction protocols,[26] deeper sequencing, high-accuracy sequencing technologies, unique molecular identifiers, and/or incorporation of low frequency somatic variant callers[27] are needed to assess if these lower-frequency variants should be included in a future HG002 mosaic benchmark version.

External validation of the benchmark confirmed that the benchmark meets GIAB's RIDE principle across a variety of somatic and mosaic variant callers. Specifically, manual curation determined that it reliably identifies both false positives and false negatives across variant calling methods and sequencing technologies.

While DNA batch effects have not previously been reported in GIAB samples when looking at germline variants, we identified substantial batch effects for mosaic variants when comparing HG002 NIST RM 8391 and non-RM (Coriell, NA24385) using mosaic VAF data. Given the higher VAFs found for mosaic benchmark variants in sequencing datasets generated using non-RM DNA, we suggest using datasets generated using NIST HG002 RM DNA to perform mosaic benchmarking.

We envision the HG002 mosaic benchmark as a GIAB somatic resource applied in use cases including, but not limited to, (1)

benchmarking mosaic variant callers, (2) acting as negative controls for either WGS somatic callers or targeted clinical sequencing in tumor-only mode, (3) benchmarking somatic variant callers in tumor-normal mode using GIAB mixtures, (4) acting as a dataset that germline researchers can use to filter low-freqeuncy somatic variants from their data, and (5) benchmarking for some types of off-target genome edits. Currently, commonly used small variant benchmarking methods, e.g., hap.py and rtg vcfeval, do not consider VAFs or properly handle differences in how low-frequency variants are represented in VCFs. Benchmarking method development and community-defined best practices will significantly improve the utility of the mosaic benchmark set and, in turn, low-frequency variant calling performance.

The Medical Device Innovation Consortium (MDIC) is a public-private partnership with the aim of advancing regulatory science for the development and assessment of medical devices. The MDIC launched the Somatic Reference Sample (SRS) Initiative to develop reference samples that can be widely distributed, so that all stakeholders can have access to the same reference samples. To meet this goal, the SRS Initiative intends to genetically engineer the well-characterized GIAB HG002 genome[4] with somatic variants. To establish a baseline for the engineered cell lines, the mosaic benchmark set generated in this study using the unedited AJ trio genomes will be used to assess and validate on- and off-target edits of clinically relevant cancer variants in HG002.

We report an HG002 mosaic variant benchmark v.1.1 generated using a trio-based framework with GIAB RM and Strelka2 to produce a high-confidence mosaic callset. This benchmark is important given growing interest from the research community in understanding somatic mosaicism, such as the NIH Common Fund project SMaHT (https://smaht.org/). This HG002 mosaic benchmark will also serve as the genomic background for the upcoming SRS Initiative, an MDIC project, focused on providing RMs to improve cancer and disease diagnostics.

## Limitations of this study

A limitation of this benchmark is the relatively small number of true mosaic variants compared to other benchmarking approaches, such as sample mixtures, tumor cell lines, and

**Table 2. External evaluation results**

| Seq. data | Cov. | Variant caller | Mode | Both | Callset only | Curated |
|---|---|---|---|---|---|---|
| Illumina | 40× | DRAGEN v.4.0.3 | T only | 21 | 1,914 | 10 |
| Illumina | 300× | TNScope v.202010 | T only | 74 | 13 | 13 |
| Illumina | 300× | TNScope v.20230801 | T/N | 79 | 20 | 20 |
| Element | 70× | DeepSomatic v.1.6.1 | T/N | 81 | 273 | 4 |
| Illumina | 300× | DRAGEN v.4.3.6 | T only | 85 | 105 | 9 |
| Illumina | 300× | DRAGEN v.4.3.6 | T only | 85 | 41 | 6 |
| Illumina | 300× | DeepSomatic v.1.6.0 | T/N | 84 | 2,530 | 28 |
| Element | 100× | DeepSomatic v.1.6.0 | T/N | 81 | 1,950 | 6 |
| Pacbio Onso | 35× | DeepSomatic v.1.6.0 | T/N | 45 | 205 | 8 |
| Pacbio Revio[a] | 130× | DeepSomatic v.1.6.0 | T/N | 83 | 1,232 | 17 |
| Illumina | 300× | DRAGEN v.4.2.4 | T only | 69 | 135 | 10 |
| Illumina | 300× | Mutect2 (GATK4) | T only | 82 | 80,965 | 10 |
| Illumina | 300× | Strelka 2.9.10 | T/N (HG003N) | 81 | 399 | 10 |
| Illumina | 300× | Strelka 2.9.10 | T/N (HG004N) | 81 | 442 | 10 |
| Illumina | 300× | NeuSomatic v.0.1.4 | T/N | 83 | 1 | 1 |
| Illumina | 300× | Consensus | T/N | 84 | 1 | 1 |

Sixteen variant callsets were provided by external collaborators to validate the draft mosaic benchmark set. The callsets were compared to the v.1.0 mosaic and v.4.2.1 small variant benchmark sets. The callsets were generated using 4 different sequencing platforms with different coverages and variant callers. The variant callers were run in tumor-only (T-only) or tumor-normal (T/N) mode. Unless indicated otherwise, combined HG003 and HG004 data were used as normal. The "both" column represents the number of variants in the v.1.0 mosaic benchmark present in the query callset. The "callset only" column indicates the number of variants in the query callset, not in the v.1.0 mosaic or v.4.2.1 small variant benchmark set. A random subset of these variants ("curated" column) was manually curated to ensure the benchmark set can be used to reliably identify false positive variant calls. Note that these comparisons were intended to evaluate the accuracy of the benchmark and not the performance of each method because the participants were not blinded, some methods were experimental, and most methods are under active development.
[a]In addition, PacBio Revio sequencing was not from the NIST RM DNA, so many callset-only variants appear to be true mosaic variants with >5% VAFs in their batch of the HG002 cell line and not in the NIST RM DNA.

spike-ins. In addition, these variants may not precisely resemble biological mosaics and somatic variants and are at a higher VAF than some biological variants. However, characterizing these variants and excluding potential mosaic variants with VAFs above 2% provides a more comprehensive negative control for identifying false positives and improves the utility of the GIAB germline benchmark for HG002. In addition, it is complementary to approaches such as sample mixtures, spike-ins, and engineering variants by providing a well-characterized background DNA sample that is commonly used in these studies.

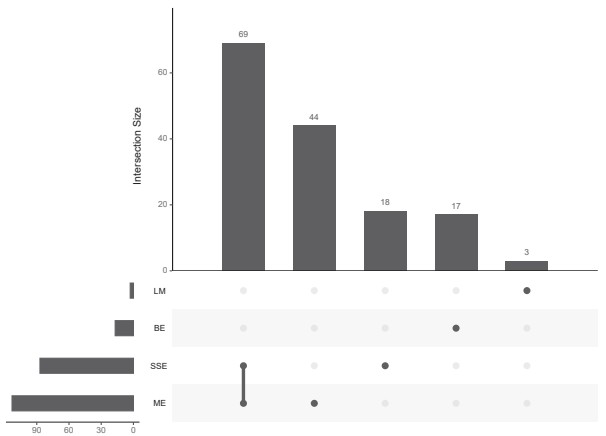

**Figure 5. Summary of external evaluation manual curations**
A total of 163 variants were curated across all external callsets. The variants were categorized based on the curator's notes as ME (mapping errors), SSE (systematic sequencing errors), BE (batch effects; sequencing data generated from non-NIST RM DNA), or LM (likely mosaic). Intersection size is the number of variants in the variant category set.

## RESOURCE AVAILABILITY

### Lead contact
Requests for further information and resources should be directed to and will be fulfilled by the lead contact, Nathan D. Olson (nolson@nist.gov).

### Materials availability
This study did not generate any new material. The HG002 DNA characterized in this study is available from NIST and Coriell.

The NIST HG002 RM DNA is available from the NIST RM website: https://shop.nist.gov/ccrz__ProductDetails?sku=8391&cclcl=en_US.

The non-RM HG002 DNA (NA24385) is available from Coriell Institute: https://catalog.coriell.org/0/Sections/Search/Sample_Detail.aspx?Ref=NA24385&Product=DNA.

### Data and code availability
The HG002 mosaic benchmark v.1.1 files and external validation VCFs can be found at the GIAB ftp site: https://ftp-trace.ncbi.nlm.nih.gov/ReferenceSamples/giab/release/AshkenazimTrio/HG002_NA24385_son/mosaic_v1.10/GRCh38/SNV.

The mosaics database used to generate the benchmark is located at https://ftp-trace.ncbi.nlm.nih.gov/ReferenceSamples/giab/release/AshkenazimTrio/HG002_NA24385_son/mosaic_v1.10/GRCh38/SNV/SupplementaryFiles/.

The MRG full BED from Wagner et al. 2022[8] is here: https://github.com/usnistgov/cmrg-benchmarkset-manuscript/blob/master/data/gene_coords/unsorted/GRCh38_mrg_full_gene.bed.

Sequencing data and data resources used to generate and characterize the mosaic benchmark set are listed in the key resources table.

The code used in this study can be accessed at the following GitHub repository: https://github.com/usnistgov/giab-HG002-mosaic-benchmark (https://doi.org/10.5281/zenodo.17633128).

## ACKNOWLEDGMENTS

The authors would like to thank Kevin Keissler, Justin Wagner, and Maryellen de Mars for feedback on the draft manuscript. C.E.M. thanks the Scientific Computing Unit (SCU), the WorldQuant and GI Research Foundations, NASA (80NSSC23K0832), the National Institutes of Health (NIH; R01ES032638 and U54AG089334), and the LLS (MCL7001-18, LLS 9238-16, and 7029-23). C.F. and ITER colleagues were supported by Instituto de Salud Carlos III (CB06/06/1088 and PI23/00980) and co-financed by the European Regional Development Funds, "A way of making Europe," from the European Union, Ministerio de Ciencia e Innovación (RTC-2017-6471-1, AEI/FEDER, UE), and Cabildo Insular de Tenerife (CGIEU0000219140 and A0000014697); by the agreement with Instituto Tecnológico y de Energías Renovables (ITER) (OA23/043) to strengthen scientific and technological education, training, research, development, and scientific innovation in genomics, epidemiological surveillance based on massive sequencing, personalized medicine, and biotechnology; and by the agreement between Consejería de Educación, Formación Profesional, Actividad Física y Deportes and Cabildo Insular de Tenerife (AC0000022149). This work, as part of the Somatic Reference Samples Initiative, is funded by grants from Illumina, Quidel, the Gordon and Betty Moore Foundation, and the National Philanthropic Trust. P.C.B., Y.P., T.N.Y., and M.F.E.M. were supported by the NIH through awards P30CA016042, U2CCA271894, and U24CA248265 and by DOD PCRP award W81XWH2210247. Certain equipment, instruments, software, or materials are identified in this paper in order to specify the experimental procedure adequately. Such identification is not intended to imply recommendation or endorsement of any product or service by NIST, nor is it intended to imply that the materials or equipment identified are necessarily the best available for the purpose.

## AUTHOR CONTRIBUTIONS

J.M.Z. and N.D.O. designed the study. J.M.Z., N.D.O., C.A.D., A.A.A., and M.H.C. performed the analyses. D.J., L.A.R.-R., A.M.-B., J.M.L.-S., C.F., B.Y., S.M.E.S., Y.W., M.R., A.V., L.M., W.-T.C., S.C., J.H., R.M., G.P., A.C., P.-C.C., K. Shafin, D.C., A.K., L.B., M.F.E.M., Y.P., T.N.Y., P.C.B., K. Sienkiewicz, J.F., C.E.M., B.R.L., C.A.R.-R., and S.K. participated in the external evaluation of the draft benchmark set. C.A.D., N.D.O., A.A.A., M.H.C., and J.M.Z. wrote the manuscript. All authors reviewed and approved the manuscript.

## DECLARATION OF INTERESTS

C.E.M. is a co-founder of Onegevity. Y.W., M.R., A.V., L.M., W.-T.C., S.C., J.H., R.M., and G.P. are Illumina employees and equity owners. A.C., P.-C.C., K. Shafin, D.C., A.K., and L.B. are employees of Google LLC and receive equity compensation. P.C.B. sits on the scientific advisory boards of Intersect Diagnostics, Inc., and BioSymetrics, Inc., and previously sat on that of Sage Bionetworks.

## DECLARATION OF GENERATIVE AI AND AI-ASSISTED TECHNOLOGIES IN THE WRITING PROCESS

During the preparation of this work, the authors used Azure OpenAI in order to revise text and code as drafting metadata, such as formatting data dictionaries. After using this tool or service, the authors reviewed and edited the content as needed and take full responsibility for the content of the publication.

## STAR★METHODS

Detailed methods are provided in the online version of this paper and include the following:

- KEY RESOURCES TABLE
- EXPERIMENTAL MODEL AND STUDY PARTICIPANT DETAILS
- METHOD DETAILS
  - Reference material sequencing datasets
  - Limit of detection assessment
  - Benchmark generation
  - Benchmark characterization and validation

## SUPPLEMENTAL INFORMATION

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

## STAR★METHODS

### KEY RESOURCES TABLE

| REAGENT or RESOURCE | SOURCE | IDENTIFIER |
|---|---|---|
| **Biological samples** | | |
| NIST Reference Material DNA, HG002 (Ashkenazi Jewish son) | NIST | RM 8391 |
| PGP Ashkenazi Jewish son NA24385 | Coriell | NA24385 |
| **Deposited data** | | |
| Illumina 2 × 148 bp WGS, HG002 (RM 8391/RM 8392), 300× | NCBI SRA; Zook et al. 2016[4] | SRA: SRX847862–SRX848005 |
| Illumina 2 × 148 bp WGS, HG003 (RM 8392), 300× | NCBI SRA; Zook et al. 2016[4] | SRA: SRX848006–SRX848173 |
| Illumina 2 × 148 bp WGS, HG004 (RM 8392), 300× | NCBI SRA; Zook et al. 2016[4] | SRA: SRX848174–SRX848317 |
| BGI 2 × 150 bp WGS, HG002 (RM 8391/RM 8392), 100× | NCBI SRA; this paper | SRA: SRX22242218 |
| Element 2x150bp, PCR-free, standard insert, HG002 (RM 8391/RM 8392), 81x | NCBI SRA, GIAB FTP, this paper | SRA: SRR30945810, https://ftp-trace.ncbi.nlm.nih.gov/ReferenceSamples/giab/data/AshkenazimTrio/HG002_NA24385_son/Element_AVITI_20231018/HG002_GRCh38-GIABv3_Element-StdInsert_2X150_81x_20231018.bam |
| Element, 2x150bp, PCR-free, long insert, HG002 (RM 8391/RM 8392), 55x | NCBI SRA, GIAB FTP, this paper | SRA: SRR30945809, https://ftp-trace.ncbi.nlm.nih.gov/ReferenceSamples/giab/data/AshkenazimTrio/HG002_NA24385_son/Element_AVITI_20231018/HG002_GRCh38-GIABv3_Element-LngInsert_2X150_55x_20231018.bam |
| PacBio HiFi Sequel 10 kb, HG002 (RM 8391/RM 8392), 30× | NCBI SRA; Wenger et al. 2019[23] | SRA: SRX5327410 |
| PacBio HiFi Sequel 15 kb, HG002 (RM 8391/RM 8392), 28× | NCBI SRA | SRA: SRX6908796, SRA: SRX6908797, SRA: SRX6908798 |
| PacBio HiFi Revio 20 kb, HG002 (RM 8391/RM 8392), 48× | GIAB FTP | https://ftp-trace.ncbi.nlm.nih.gov/ReferenceSamples/giab/data/AshkenazimTrio/HG002_NA24385_son/PacBio_HiFi-Revio_20231031/HG002_PacBio-HiFi-Revio_20231031_48x_GRCh38-GIABv3.bam |
| Element, 2x150bp, standard insert, HG002 (NA24385), 100X | Arslan et al. 2024[28] | SRA: SRX17079410 |
| PacBio CCS/HiFi (32X), Sequel II, 11kb | NCBI SRA | SRA: SRX5527202 |
| PacBio CCS/HiFi Sequel II merged 15 and 20kb smrtcells, HG002 (NA24385), 52x | | 15 kb - SRA: SRX7083054, SRA: SRX7083055, SRA: SRX7083056, SRA: SRX7083057; 20 kb - SRA: SRX7083058, SRA: SRX7083059 |
| PacBio CCS/HiFi Revio, 15-20kb, 30X | PacBio website | https://downloads.pacbcloud.com/public/revio/2022Q4/ |
| GIAB stratifications used for benchmarking | Dwarshuis et al. 2024[29] | https://ftp-trace.ncbi.nlm.nih.gov/ReferenceSamples/giab/release/genome-stratifications/v3.3/ |
| v4.2.1 Benchmark VCF and BED files | Wagner et al. 2022[7] | https://ftp-trace.ncbi.nlm.nih.gov/ReferenceSamples/giab/release/ |

*(Continued on next page)*

*Continued*

| REAGENT or RESOURCE | SOURCE | IDENTIFIER |
|---|---|---|
| **Experimental models: Cell lines** | | |
| Son of Ashkenazi Jewish ancestry (HG002) NIST | NIST Office of Reference Materials; Coriell/NIGMS; PGP | NIST RM8391/RM8392; GM24385; RRID: CVCL_1C78 |
| Father of Ashkenazi Jewish ancestry (HG003) | NIST Office of Reference Materials; Coriell/NIGMS; PGP | NIST RM8392; GM24149; RRID: CVCL_1C54 |
| Mother of Ashkenazi Jewish ancestry (HG004) | NIST Office of Reference Materials; Coriell/NIGMS; PGP | NIST RM8392; GM24143; RRID: CVCL_1C48 |
| **Software and algorithms** | | |
| Code used to generate benchmark | https://doi.org/10.5281/zenodo.17633128 | https://github.com/usnistgov/giab-HG002-mosaic-benchmark# |
| NovoAlign v3.02.07 | Novocraft | http://www.novocraft.com |
| SAMtools v1.5, v1.9, v1.12, v1.13, v1.15, v1.16, v1.18 | Li et al. 2009,[30] Danecek et al. 2021[31] | http://www.htslib.org |
| BCFtools v1.15, v1.17 | Danecek et al. 2021[31] | http://www.htslib.org |
| BWA-MEM v0.7.15, v0.7.17 | Li & Durbin 2009[32] | https://github.com/lh3/bwa |
| Strelka2 v2.8.4, v2.9.5, v2.9.10 | Kim, S. et al. 2018[33] | https://github.com/Illumina/strelka |
| hap.py v0.3.14 | Krusche et al. 2019[12] | https://github.com/Illumina/hap.py |
| vcfeval v3.11 | Cleary et al. 2015[34] | https://github.com/RealTimeGenomics/rtg-tools |
| Bam-readcount v1.0.1 | Khanna et al. 2021[28] | https://github.com/genome/bam-readcount |
| binconf (R package) | A. Agresti and B.A. Coull 1998[35] | https://cran.r-project.org/package=Hmisc |
| WhatsHap v0.7 | Patterson et al. 2015[36] | https://whatshap.readthedocs.io |
| pbmm2 | Pacific Biosciences | https://github.com/PacificBiosciences/pbmm2 |
| PacBio HiFi-human-WGS-WDL pipeline | Pacific Biosciences | https://github.com/PacificBiosciences/HiFi-human-WGS-WDL |
| glmmTMB (R package) | Brooks et al. 2025[37] | https://cran.r-project.org/package=glmmTMB |
| Seqtk v1.3 | | https://github.com/lh3/seqtk |
| precisionFDA hap.py app | precisionFDA | https://precision.fda.gov/apps/app-F5YXbp80PBYFP059656gYxXQ-1 |
| DNAnexus strelka2 app v2.9.10 | DNAnexus | https://www.dnanexus.com/ |
| Bedtools v2.30 | Quinlan and Hall 2010[38] Quinlan et al. 2014[39] | https://github.com/arq5x/bedtools2 |
| Rstudio v2023.6.1.524 | | http://www.posit.co/ |
| IGV v2.16.2 | Robinson et al. 2011; Robinson et al. 2017 | https://igv.org/doc/desktop/ |
| Sentieon v202010, v2023.08.0 | Sentieon | https://support.sentieon.com/docs/ |
| DRAGEN v4.0.3, v4.2.4, v.4.3.6 | Illumina, Behera et al. 2025[40] | https://support.illumina.com/sequencing/sequencing_software/dragen-bio-it-platform.html |
| DeepSomatic v1.6.1 | Park et al. 2024[41] | docker.io/google/deepsomatic:1.6.1 |
| BWA, v0.7.15-r1188 | Li and Durbin 2009 | https://github.com/lh3/bwa |
| GATK4 v3.7.0, v4.2.0.0 | DePristo et al. 2011[42] | https://gatk.broadinstitute.org/hc/en-us |
| RePlow, v1.1.0 | Kim et al. 2019 | https://sourceforge.net/projects/replow/ |
| Picard v2.18.0, v3.0.0 | Broad Institute | https://broadinstitute.github.io/picard/ |
| NeuSomatic v0.2.1 | Sahraeian et al. 2019[43] | https://github.com/bioinform/neusomatic |
| Mutect2 v4.4.0.0 | Cibulskis, K. et al. 2013[44] | https://gatk.broadinstitute.or |
| SomaticSniper(1.0.5.0) | Larson, D. E. et al. 2011[45] | https://github.com/genome/somatic-sniper |
| MuSE v1.0rc, v2.0.4 | Fan, Y. et al. 2016[46] | https://github.com/wwylab/MuSE |

*(Continued on next page)*

*Continued*

| REAGENT or RESOURCE | SOURCE | IDENTIFIER |
|---|---|---|
| VarDict (v1.5.1) | Lai, Z. et al. 2016[47] | https://github.com/AstraZeneca-NGS/VarDict |
| PipeVal (v4.0.0-rc.2) | Patel et al. 2024[48] | https://github.com/uclahs-cds/package-PipeVal |
| Base Quality Score Recalibration (BQSR) (v4.2.4.1) | McKenna et al. 2010[49] | https://gatk.broadinstitute.or |
| Manta v1.6.0 | Chen et al. 2016[50] | https://github.com/Illumina/manta |

## EXPERIMENTAL MODEL AND STUDY PARTICIPANT DETAILS

Genomic DNA for Genome in a Bottle Ashkenazi Jewish trio (HG002: son, HG003: father, HG004: mother) from the NIST reference material RM8391 and RM8392 as well as cells (GM2385) and genomic DNA (NA2385) from Coriell (GM2385) were used in this study. This trio is from the Personal Genome Project and have a broad consent permitting commercial redistribution and recontacting participants for further sample collection in addition to making their genome sequence publicly available. The cell lines and DNA are derived from lymphoblastoid cells and immortalized using EBV.

## METHOD DETAILS

### Reference material sequencing datasets

High coverage (>100x) whole genome sequencing datasets were generated using three different short-read and one long-read sequencing platforms. High-coverage (300x) Illumina short-read data, previously described,[4] was used to identify potential mosaic variants. Briefly, six vials of NIST reference gDNA from the GIAB Ashkenazi Jewish (AJ) trio cell lines (HG002-son, HG003-father, HG004-mother; Figure 1A) were extracted and Illumina TruSeq PCR-free kit was used to generate paired-end libraries. DNA from each replicate library was multiplexed and sequenced with 2x148bp on an Illumina HiSeq2500 v1 in Rapid mode at NIST (Gaithersburg, MD). Pooled reads from each GIAB sample were mapped to GRCh38 genome (GCA_000001405.15, hs38d1) using NovoAlign v3.02.07 and BAM files were generated, sorted, and indexed with SAMtools v0.1.18. High coverage reads (300x) from the GIAB AJ Trio (NCBI Biosample SAMN03283347, SAMN03283345, SAMN03283346; PGP huAA53E0, hu6E4515, hu8E87A9) were retained for subsequent processing in this study (Table 1; Figure 1A).

Additional high-coverage HG002 orthogonal datasets were used for mosaic variant validation: BGI (100x), Element (standard and long insert - 136X total), and PacBio HiFi (Sequel and Revio- 106x total). High coverage PCR-free 2x150bp BGI data was generated by BGI using the DNBSEQ sequencing platform and basecalled using the DNBSEQ basecalling software with default parameters. Reads were aligned to the hg38 genome (https://hgdownload.cse.ucsc.edu/goldenPath/hg38/bigZips/latest/hg38.fa.gz) using bwa-mem v0.7.17. For Element, standard (400bp) and long insert (1.3 kb) libraries were prepared with 1µg gDNA of HG002 NIST RM, the Kapa Hyper Prep Kit, and Kapa UDI. Seven rounds of Covaris shearing were performed on the DNA libraries, which were subsequently treated with USER enzyme and circularized with Adept Rapid PCR Free workflow using the Element Library Compatibility Kit v1. In a separate run for each insert size, 2x150bp libraries were sequenced across both flow cell lanes on the Element Biosciences AVITIplatform. Reads from each run were mapped to GRCh38-GIABv3 (https://ftp-trace.ncbi.nlm.nih.gov/ReferenceSamples/giab/data/AshkenazimTrio/HG002_NA24385_son/Element_AVITI_20231018/) using bwa-mem v0.7.17, and SAMtools v1.18 was used to assess mapping results for each insert size. All PacBio libraries were created and sequenced at Pacific Biosciences. Two libraries were previously generated,[23] which sheared HG002 NIST RM gDNA using a Megaruptor with the 20 kb protocol. Size selection was performed on a sageELF DNA system (Sage Science) to capture 10kb and 15kb bands for sequencing. Band sizes were verified on an Agilent 2100 BioAnalyzer using a DNA 12000 kit used as input into the SMRTbell prep kit 3.0, and sequenced to 30x and 28x-fold coverage, respectively, on a PacBio Sequel System. Default parameters were used to generate consensus (CCS) reads. Using pbmm2 with CCS presets, reads were subsequently aligned to GRCh38_no_alt_analysis reference and phased with WhatsHap v0.7. A PacBio Revio library was created (unpublished, 2023) by shearing 6µg HG002 NIST RM gDNA with a Megaruptor 3 and using 4.5µg as input for the SMRTbell prep kit 3.0. PippinHT size selection followed along with a 10kb cut. PacBio Revio polymerase and sequencing kits were used for sample prep for sequencing to 48x-fold coverage at Pacific Biosciences, and two SMRT cells were subsequently run for the HG002 sample (24-h movies). Using the PacBio HiFi-human-WGS-WDL pipeline (https://github.com/PacificBiosciences/HiFi-human-WGS-WDL), HiFi reads were aligned independently to GRCh38-GIABv3 (https://ftp-trace.ncbi.nlm.nih.gov/ReferenceSamples/giab/data/AshkenazimTrio/HG002_NA24385_son/PacBio_HiFi-Revio_20231031/).

### Limit of detection assessment

A minimum VAF of 5% for variants to include in the mosaic benchmark set was identified using in-silico mixtures of HG002 and HG003. AJ trio Illumina WGS 300x datasets were subset to chromosome 20 and downsampled using SAMtools (v1.9) to create

simulated HG002 + HG003 samples with six different allele fractions (AFs), HG002 unmixed (50% AF), 50% HG002 + 50% HG003 (25% AF), 20% HG002 + 80% HG003 (10% AF), 10% HG002 + 90% HG003 (5% AF), 2% HG002 + 98% HG003 (1% AF), and HG003 unmixed (0% AF). Variant calling was performed with the Strelka2 app on DNAnexus (Strelka v2.8.4 and SAMtools v1.5) using hs37d5 as the reference. Callsets were benchmarked with hap.py in the precisionFDA app and analyzed to ascertain a limit of detection (LOD; Figure 1B). Recall was observed for all SNVs at 99% and 97% for passing SNVs down to 5% VAF, and respectively, fell to 25% and 8% approaching 1% VAF (Figure S2). Based on these preliminary results, we targeted variants with VAFs >5% and <30% for inclusion in the HG002 mosaic benchmark set. The GIAB v4.2.1 small variant benchmark only includes variants down to 30% VAF.

### Benchmark generation

For the HG002 mosaic benchmark set generation, an initial set of mosaic variants was identified using the Strelka2 somatic variant caller and characterizable genomic regions as the intersection of the GIAB HG002, HG003, and HG004 v4.2.1 benchmark regions, excluding complex (i.e., nearby) variants in any sample. Complex variants were excluded as they tend to cause errors due to differences in variant representation. These initial variants and genomic regions were refined based on a set of heuristics defined based on observations from manually curating targeted subsets of the initial mosaic variants.

The initial set of low fraction variants in HG002 (son) were identified with the Strelka2 somatic variant caller in the Strelka2 DNA-nexus app (Strelka v2.9.10 and SAMtools v1.13) and AJ trio 300x WGS data (Figure 1C; Table S1) using the HG002 BAM as tumor, a combined parental BAM (HG003 + HG004) as normal, and the GRCh38 reference including the hs38d1 decoy FASTA, where the job run was split by chromosome. SNV and indel VCFs were concatenated into a single VCF. Variants with a `FILTER` column value of `VARIANT_DETECTED_IN_NORMAL` (i.e., HG003 and HG004) were removed using bcftools v1.16, while variants not detected in the normal sample with the Strelka2 tumor allele read (TAR) depth value >5 were retained. The merged VCF was formatted with custom scripts for downstream analyses.

Next, variants in the GIAB HG002 v4.2.1 small variant benchmark were excluded from the initial set of low fraction variants. The reformatted Strelka2 VCF compared to the v4.2.1 benchmark set using vcfeval v3.11 with the –squash-ploidy flag and the intersection of the HG002, HG003, and HG004 v4.2.1 benchmark regions as target regions. The resulting set of low VAF variants were further refined using custom scripts to exclude regions with GIAB AJ trio v4.2.1 complex and structural variants (see Figure S2). From the resulting set of low VAF variants, Strelka2 passing variants are referred to as candidates and non-passing variants as putative mosaic variants.

A database of candidate and potential mosaic variants was generated for the development and application of a set of heuristics used to define the final mosaic benchmark set. The mosaic variants database (Data Availability) contained variant information from the Strelka2 VCF, variant support from multiple orthogonal sequencing datasets, and genomic context annotations. Bam-read-count[51] was used to calculate reference and alternate allele read support for HG002 Illumina (300x) and orthogonal HG002-GRCh38 BAMs from three WGS datasets: BGI (100x), Element (standard and long insert - 136x total), and PacBio HiFi (Sequel and Revio: 106x total). To remove low quality variants, base and mapping quality filters (-b25, -q40) and a minimum threshold of ≥2 read support per orthogonal tech for each variant was applied. The read support counts were used to calculate overall VAF estimates and confidence intervals per orthogonal dataset. Binomial confidence intervals (CI) were calculated using binconf R package.[35,52,53] Independent orthogonal CIs used a one-sided confidence coefficient of 95%, while the combined orthogonal CI used a one-sided confidence coefficient of 99%. GIAB GRCh38 genome stratifications were used to annotate variants in the database with genomic context using bcftools v1.15.

A series of heuristics (see Figure S3) was used to identify variants for manual curation. This process used combined orthogonal CIs, PacBio read depth thresholds, removed variants that overlap germline indels, and partitioned the data by genomic context. To be considered for manual curation, we required a ≥0.01 (using a 99% one-sided confidence interval) lower CI threshold of the VAF from the combined orthogonal technologies for easy-to-map variants and a ≥0.05 (using a 95% one-sided confidence interval) lower PacBio CI threshold for variants in difficult-to-map regions.

The HG002 mosaic SNV benchmark was defined using the following heuristics in a decision tree (Figure S3). Initial filtering of the mosaic database (366,728) removed variants with a combined orthogonal upper CI ≤ 0.03. Of the passing variants (6,178), those that fell below the 0.5% quantile thresholds for either total number of reads across combined orthogonal methods or PacBio depth were filtered. Next, database variants that overlapped germline indels were removed. Variants with $X_{ci}/n_{ci}$ (i.e., total number of variant reads across combined orthogonal methods/total number of reads across combined orthogonal methods) > 0.5 were removed. Database variants that remained were then partitioned into easy-to-map and not easy-to-map (i.e., low mappability) bins using GRCh38_notinlowmappabilityall.bed.gz from the GIAB v3.1 stratifications.[29] A combined orthogonal lower CI ≥ 0.05 filter was applied to the easy-to-map variants (2,323), with 121 passing and were retained for manual curation. Not easy-to-map variants (1,668) were partitioned into homopolymer and non-homopolymer bins using GRCh38_AllHomopolymers_gt6bp_imperfectgt10bp_slop5.bed from the GIAB v3.1 stratifications. After applying a PacBio lower CI ≥ 0.05 threshold, 14 not easy-to-map, non-homopolymer variants passed, and totaled 135 mosaic database variants for manual curation (see Figure S3: green boxes). Variants that did not adhere to decision tree heuristics were excluded from the benchmark VCF. If they were likely false positives or <2% VAF. Variants were excluded from both VCF and BED files if the evidence was unclear or the VAF could not be confidently determined to be <0.02 or ≥0.05 (see Figure S3: red, yellow boxes).

## Benchmark characterization and validation

HG002 mosaic benchmark variants in medically relevant genes were identified by intersecting the benchmark VCF with a BED file containing coordinates for genes in a previously generated curated list of 5,026 medical relevant genes (MRGs).[8] MRG coverage info and the number of bases occurring in HG002 mosaic benchmark regions were obtained by comparing the mosaic benchmark BED and GRCh38 full MRG BED files.

Differences in variant allele frequencies for the 85 mosaic benchmark set variants were found between datasets generated using HG002 NIST RM DNA (RM) and HG002 DNA from other sources (non-RM). Bam-readcount data for Element (standard insert libraries) and PacBio HiFi (Revio) were used to get counts for reads supporting the mosaic variant and reference base. We tested for global and individual variant allele frequencies between DNA sources (RM vs. non-RM) using a generalized linear mixed-effects model (GLMM). We modeled the number of alternate reads (`alt_count`) out of total reads (`alt_count + ref_count`) per variant per sample using a binomial distribution. The model included DNA source (`dna_source`) and sequencing platform (`platform`) as fixed effects, and variant ID (`variant_id`) as a random intercept to account for baseline allele frequency heterogeneity. The model was fit using the glmmTMB package[37] in R. A likelihood ratio test (LRT) was used to compare the full model with a reduced model excluding the `dna_source` term to test whether VAF profiles differed between RM and nonRM sources. To identify individual variants with significant VAF differences between RM and nonRM samples, we fit a separate binomial generalized linear model (GLM) for each variant. We extracted the log-odds estimate for `dna_source = RM`, its 95% confidence interval, and the associated $p$-value. $p$-values were adjusted across all variants using the Benjamini–Hochberg procedure to control the false discovery rate (FDR).

The draft mosaic benchmark set was shared with eight groups working on somatic or mosaic variant calling methods for external validation. Each group compared their somatic or mosaic callset(s) against a draft version of the HG002 mosaic benchmark to determine if the benchmark set can be used to reliably identify errors.[11] Tumor-normal and/or tumor-only mode(s) were used to generate callsets with commercial and open-source variant calling tools. Briefly, external collaborators used publicly available data generated using the HG002, HG003, and HG004 NIST reference material DNA: Illumina, PacBio HiFi, Element, Ultima, or Onso (non-RM DNA). HG002 was used as the tumor sample, and groups that performed tumor-normal sequencing used either HG003, combined HG003 and HG004, or HG004 as the normal sample. The eight groups used different variant callers, including Illumina's DRAGEN pipeline, the deep-learning model DeepSomatic,[41] Sentieon TNscope,[54] and widely used heuristic or probabilistic tools such as Mutect2,[55] Strelka2,[33] VarDict,[47] and Lancet.[56] Detailed descriptions of the methods used by the eight groups provided below. Using the hap.py[57] benchmarking tool with vcfeval[34] option enabled, the external callsets were compared to the HG002 mosaic benchmark v1.0 and also a combined HG002 mosaic benchmark v1.0 plus GIABv4.2.1 HG002 germline small variant benchmark VCF to assess if the benchmark could reliably identify false positives. A subset of the discrepancies with VAFs between 5 and 30% was manually curated to validate the benchmark (Table S5).

### *External validation method details*

Detailed methods provided by groups during the external evaluation are provided below. Text was formatted for publication but left unedited.

*Children's Mercy Kansas City.* Input Data: For this benchmark, we used Illumina PCR-free whole genome (WGS) data 2x150bp 40X per individual FASTQ files sequenced at Children's Mercy Kansas City.

Read Alignment: DRAGEN 4.0.3 performed the alignment on the multi genome graph reference available here: https://support.illumina.com/downloads/dragen-reference-genomes-hg38.html.

Alignments and Mark duplicates: dragen --config alignment.cfg
alignment.cfg file

```
enable-map-align = true
enable-map-align-output = true
enable-bam-indexing = true
enable-sort = true
enable-deterministic-sort = true
enable-duplicate-marking = true
remove-duplicates = false
ref-dir = hg38+alt_masked+cnv+graph+hla+rna-8-r2.0-1.
```

Variant Calling: DRAGEN 4.0.3 somatic variant calling in tumor only mode detects the mosaic variant calls, dragen --config somatic-variant-calling.cfg --tumor-bam-input alignment.bam.

somatic-variant-calling.cfg file

```
enable-map-align = false
enable-map-align-output = false
enable-bam-indexing = false
enable-sort = true
enable-deterministic-sort = true
enable-duplicate-marking = false
remove-duplicates = false
ref-dir = hg38+alt_masked+cnv+graph+hla+rna-8-r2.0-1
enable-variant-caller = true
```

```
    vc-hard-filter = DRAGENHardSNP:snp: MQ < 30.0 || MQRankSum < −12.5 ||
ReadPosRankSum < −8.0;DRAGENHardINDEL:indel: ReadPosRankSum < −20.0
    vc-max-alternate-alleles = 6
    vc-target-coverage = 2000
    vc-min-read-qual = 20
    vc-min-base-qual = 10
    vc-min-call-qual = 20.0
    vc-min-reads-per-start-pos = 5
    vc-emit-zero-coverage-intervals = true
    vc-decoy-contigs = chrEBV
    vc-decoy-contigs = hs38d1
    enable-smn = false
    enable-cyp2d6 = false
    enable-hrd = false
    vc-ml-enable-recalibration = true.
```

Tool Versions: Illumina DRAGEN Bio-IT Platform v4.0.3.

Curation Decisions: Finally, using the "MOSAIC >5% VAF?" column from the mosaic benchmark curation sheet, we declared: If our procedure detected the variant, our verdict was TRUE, because our data is relatively low depth and MOSAIC >5% VAF is required to be called. If our procedure didn't detect the variant, our verdict was FALSE. We performed a manual curation with IGV to complement our verdict in the cases of discordance between our verdict and the GIAB call.

*Cornell (Mason lab).* Input Data: The Illumina PCR-free WGS (2x150bp, 300X) FASTQ files for sample HG002 were retrieved based on indexes from NIST's GIAB GitHub repository. The reference genome file GRCh38-GIABv3 version was retrieved from the GIAB FTP repository. A set of indexes was created utilizing bwa-mem2, GATK CreateSequenceDictionary, and samtools. The known SNP and indels references for GRCh38 assembly were downloaded from the GATK Public Resource Bundle.

Read Alignment: Each replicate from each run was processed using Sentieon's TNscope DNAseq workflow with the following steps:
Read alignments

```
    readgroup = @RG\\tID:${sample}\\tSM:${sample}\\tPL:ILLUMINA
    reference = GRCh38_GIABv3_no_alt_analysis_set_maskedGRC_decoys_MAP2K3_KMT2C_KCNJ18.fasta
    sentieon bwa mem -M -R "${readgroup}" -t $CORES \
      -K 10000000 $reference $fastqR1 $fastqR2 \
      | sentieon util sort -r $reference \
        -o ${sample}.bam -t $CORES --sam2bam -i -
```

Collect alignment metrics

```
    bam = ${sample}.bam
    sentieon driver -r $reference -t $CORES -i $bam \
      --algo MeanQualityByCycle MQmetrics_$sample.txt \
      --algo QualDistribution QDmetrics_$sample.txt \
      --algo GCBias --summary GCsummary_$sample.txt GCmetrics_$sample.txt \
      --algo AlignmentStat ALNmetrics_$sample.txt \
      --algo InsertSizeMetricAlgo ISmetrics_$sample.txt
    sentieon plot GCBias -o GC_$sample.pdf GCmetrics_$sample.txt
    sentieon plot QualDistribution -o QD_$sample.pdf QDmetrics_$sample.txt
    sentieon plot MeanQualityByCycle -o MQ_$sample.pdf MQmetrics_$sample.txt
    sentieon plot InsertSizeMetricAlgo -o IS_$sample.pdf ISmetrics_$sample.txt.
```

Mark and remove sequencing duplicates, followed by updating read tags

```
    sentieon driver -t $CORES -i $bam --algo LocusCollector --fun score_info ${sample}_score.txt
      sentieon driver -t $CORES -i $bam --algo Dedup --score_info ${sample}_score.txt --metrics
      DEDUPmetrics_$sample.txt ${sample}.md.bam
      sentieon driver -r $reference -t $CORES -i ${sample}.md.bam --algo CoverageMetrics
      DEDUPcovmetrics_$sample.
```

Base quality score recalibration

```
    known_snps = Homo_sapiens_assembly38.dbsnp138.vcf.gz
    known_indels = Mills_and_1000G_gold_standard.indels.hg38.vcf.gz
    sentieon driver -r $reference -t $CORES -i ${sample}.merged.bam --algo QualCal -k $known_snps -k
$known_indels RECAL_$sample.table
    sentieon driver -r $reference -t $CORES -i ${sample}.merged.bam -q RECAL_$sample.table \
    --algo QualCal -k $known_snps -k $known_indels RECAL_$sample.table.post
    sentieon driver -t $CORES --algo QualCal --plot --before RECAL_$sample.table \
```

```
--after RECAL_$sample.table.post RECALdiff_$sample.csv
sentieon plot QualCal -o RECAL_$sample.pdf RECALdiff_$sample.csv.
```
Final BAM files per run for each sample were merged using bamtools and indexed using samtools:
```
ls *.md.bam > bam_list.txt
bamtools merge -list bam_list.txt -out HG002.merged.bam
samtools index -@ $CORES HG002.merged.bam HG002.merged.bam.bai.
```
Update the read tags in the merged BAM file:
```
samtools view -H HG002.merged.bam | grep -v '@RG' > header.sam
  samtools view -H HG002.merged.bam | grep '^@RG' | sed -e "s/SM:2/SM:HG002--/" | awk -F'--' '{print
  $1}' ≫ header.sam
  samtools reheader header.sam HG002.merged.bam > HG002.merged.rh.bam.
```
Tool Versions: bamtools v2.5.2, GATK CreateSequenceDictionary, samtools v1.9, Sentieon v202010.
Variant Calling:
```
sentieon driver -r $reference -t $CORES -i HG002.merged.rh.bam -q RECAL_HG002.table \
  --algo TNscope --disable_detector sv --trim_soft_clip --tumor_sample "HG002" \
  -q RECAL_HG002.table --dbsnp $known_snps HG002.somatic.vcf.gz.
```
Curation Decisions: Finally, using the KEEP/?/REMOVE column from the mosaic benchmark curation sheet, we declared: If our procedure defined a variant as TP and it was marked as KEEP, our verdict was to KEEP it. If our procedure defined a variant as FP and it was marked as REMOVE, our verdict was to REMOVE it. Otherwise, we perform a manual curation with IGV to complement our verdict in the cases of discordance between our verdict and the GIAB call.

*DRAGEN.* In DRAGEN v4.3 we added mosaic detection within the DRAGEN germline pipeline, using an advanced machine learning (ML) model to detect SNP and indel mosaic variants. Our mosaic detection algorithm exploits the DRAGEN pangenome reference to recover low allele frequency (AF) calls, without requiring matched controls. Mosaic calls are integrated into a standard VCF output alongside germline variants, using tags to ease interpretation.

We present both the default DRAGEN-ML germline workflow and the enhanced DRAGEN-ML workflow with mosaic variant detection enabled. Users can enable/disable mosaic detection in the germline workflow as desired. Our mosaic detection workflow achieves remarkable recall and accuracy through three key enhancements: We improve sensitivity by recovering reads in low-mappability regions using the DRAGEN pangenome reference and associated advanced alignment algorithms. We extract active regions with a lower evidence threshold. This allows positions with lower read evidence to progress through the pipeline increasing sensitivity. We use an ML model that is trained specifically to identify mosaic variants improving specificity. The model runs after the germline pipeline has identified putative germline variants and identifies lower-AF mosaic calls in the remaining variant candidates.

The mosaic model is trained using supervised learning. Due to a shortage of authentic & validated mosaic data, we use Bamsurgeon to simulate mosaic variants (both SNP and INDEL). ~50k and ~10k mosaic variants are generated in GIAB v4.2.1 truth bed reference-homozygous positions for WGS and WES data respectively. The mosaic variant AF follows a uniform distribution ranging from 1% to 45%. We simulate mosaic variants in a range of sequencing platforms and configurations so that the model generalizes well across different sequencers, depths, lab-preparation flows, coverages, etc. We test our model using real mosaic data, admixture datasets, and reference datasets.

We train the mosaic model using rich read level features including statistical descriptions of mapping quality, base quality, strand bias, variant length, GC bias, depth, AF as well as internal HMM scores including foreign read probabilities, SSE triggers, base quality, and other statistics from VC internal processing. These features are extracted during DRAGEN variant calling. The features are used to build a model using offline training, outside the DRAGEN pipeline. The model uses a gradient-boosted ensemble of weak decision tree learners to identify mosaic variants, resulting in a very efficient and accurate model (adds only a few minutes to variant calling time without requiring hardware acceleration).

Mosaic variants are output in the same VCF file as germline variants. For a called mosaic variant, we tag the record's INFO field using a MOSAIC tag, and we set genotype (GT) to 0/1. We update the QUAL field with a confidence score calculated from the model probability output. We calibrated the mosaic pass threshold on the QUAL field to recover high confidence mosaic events based on validated mosaic data.

Input Data: For this benchmark evaluation, we used the GIAB HG002 Illumina PCR-free WGS (2x150bp, 300X) FASTQs.
Alignment and Variant Calling:
```
dragen \
  --fastq-list=<path-to-hg002_300x_fastq-list> \
  --ref-dir=<path-to-hg38-alt_masked.graph.cnv.hla.rna_v3> \
  --output-file-prefix = HG002_HiSeq_300x \
  --output-directory=<path-to-output-directory> \
  --events-log-file dragen_events.csv \
  --vc-enable-mosaic-detection = true \
  --generate-sa-tags = true \
```

```
--enable-vcf-compression = true \
--enable-variant-caller = true \
--enable-map-align = true \
--enable-map-align-output = true \
--enable-sort = true \
--enable-duplicate-marking = true \
--enable-bam-indexing = true.
```

Mosaic variants are tagged with MOSAIC vcf INFO tag and can be filtered with bcftools. We used bcftools to generate a mosaic-only hard-filtered VCF file using the following command.

```
bcftools filter -i "(INFO/MOSAIC = = 1)" HG002_300x.hard-filtered.vcf.gz -Oz \
  > HG002_300x.mosaic-only.hard-filtered.vcf.gz.
```

Tool Versions: DRAGEN v4.3.6.

### Element Biosciences

Input Data: For this benchmark evaluation, we used the GIAB AJ trio (HG002/HG003/HG004) NIST RM and sequenced using our new Element UltraQ (Q50) chemistry. The VCF was derived from 70x tumor/normal data, with 70x HG002 as the "tumor" and a 70x HG003+HG004 synthetic mix as the "normal".

Read Alignment: Each sample was processed using the following steps:

Alignment - Sentieon BWA (Sentieon-v2023.08.0) aligned to the standard hg38 (Homo_sapiens_assembly38) reference, and using the DNAscopeElementBioWGS2.0.bundle/bwa.model model.

```
sentieon bwa mem \
  -x DNAscopeElementBioWGS2.0.bundle/bwa.model \
  -M -R "@RG\tID:MAXQ-0216__GAT-APP-C138\tSM:GAT-APP-C138\tPL:ELEMENT" \
  -t 68 \
  -K 10000000 \
  $INDEX \
  GAT-APP-C138_FQD-2x150x150-70x_R1.fastq.gz GAT-APP-C138_FQD-2x150x150-70x_R2.fastq.gz \
  | sentieon util sort $bam_option -r Homo_sapiens_assembly38.fa -o GAT-APP-C138__MAXQ-0216.bam -t 68
--sam2bam -i -
```

Duplicates - marking and removal via Sentieon LocusCollector & Dedup (Sentieon-v2023.08.0)

```
sentieon driver \
  -t 24 \
  -i GAT-APP-C138__MAXQ-0216.bam \
  --algo LocusCollector \
  --fun score_info \
  GAT-APP-C138.score.txt
sentieon driver \
  -t 24 \
  -i GAT-APP-C138__MAXQ-0216.bam \
  --algo Dedup \
  --rmdup \
  --score_info GAT-APP-C138.score.txt \
  --optical_dup_pix_dist 100 \
  --metrics GAT-APP-C138.dedup_metrics.txt \
  GAT-APP-C138.deduped.bam.
```

Variant Calling - Google Deepsomatic v1.6.1 (docker.io/google/deepsomatic:1.6.1). A 70x downsampling of HG002 was used as the "tumor reads". A 70x downsampling of a synthetic HG003/HG004 was used as the "normal reads". To generate the 70x HG003/HG004, we synthetically merged a 35x HG003 and a 35x HG004 from two existing runs (MAXQ-0188 and MAXQ-0189).

```
run_deepsomatic \
  --model_type WGS \
  --ref Homo_sapiens_assembly38.fa \
  --reads_normal = GAT-APP-C140__GAT-APP-C142.deduped.bam \
  --reads_tumor = GAT-APP-C138.deduped.bam \
  --output_vcf = GAT-APP-C138.deepsomatic.output.vcf.gz \
  --sample_name_tumor = GAT-APP-C138 \
  --sample_name_normal = GAT-APP-C140__GAT-APP-C142 \
  --num_shards 68 \
  --intermediate_results_dir/tmp/intermediate_results_dir
```

Tool Versions: Sentieon v2023.08.0, Deepsomatic v1.6.1.

Curation Decisions: Finally, using the KEEP/?/REMOVE column from the mosaic benchmark curation sheet, we declared: If our procedure defined a variant as TP and it was marked as TP, our verdict was to KEEP it. If our procedure defined a variant as FP and it was marked as FP, our verdict was to REMOVE it. Otherwise, we perform a manual curation with IGV to complement our verdict in the cases of discordance between our verdict and the GIAB call.

*Google research genomics team.* Input Data: PacBio Revio 130X, Ilumina 300X, Element Cloudbreak data (~100x) from https://www.biorxiv.org/content/10.1101/2023.08.11.553043v1 (https://storage.mtls.cloud.google.com/brain-genomics-public/research/element/cloudbreak_wgs/HG002.element.cloudbreak.500bp_ins.grch38.bam), Onso data from PacBio downloads (https://downloads.pacbcloud.com/public/onso/2023Q3/WGS/hg002_30x_WGS/).

Read Alignment: Short reads aligned using BWA MEM and long reads using Minimap2. Replicates per run from the same sample were merged using MergeSamFiles from GATK. Final BAM files per run for each sample were merged using MarkDuplicates from GATK.

Tool Versions: BWA-MEM, DeepSomatic v1.6, Minimap2.

Variant Calling:

Tumor-normal or tumor-only calling.

```
   Illumina, Element, Onso
INPUT_DIR = ${PWD}/input.
OUTPUT_DIR = ${PWD}/output.
BIN_VERSION = 1.6.0
sudo docker run \
    -v ${INPUT_DIR}:${INPUT_DIR}/\
    -v ${OUTPUT_DIR}:${OUTPUT_DIR}/\
    google/deepsomatic:"${BIN_VERSION}" \
    run_deepsomatic \
    --model_type = WGS \
    --ref = ${INPUT_DIR}/GRCh38.no_alt_analysis_set.fa.gz \
    --reads_normal = ${INPUT_DIR}/${NORMAL} \
    --reads_tumor = ${INPUT_DIR}/${TUMOR} \
    --output_vcf = ${OUTPUT_DIR}/${VCF} \
    --sample_name_tumor = "tumor" \
    --sample_name_normal = "normal" \
    --num_shards = $(nproc)
PacBio.
INPUT_DIR = ${PWD}/input.
OUTPUT_DIR = ${PWD}/output.
BIN_VERSION = 1.6.0
sudo docker run \
    -v ${INPUT_DIR}:${INPUT_DIR}/\
    -v ${OUTPUT_DIR}:${OUTPUT_DIR}/\
    google/deepsomatic:"${BIN_VERSION}" \
    run_deepsomatic \
    --model_type = PACBIO \
    --ref = ${INPUT_DIR}/GRCh38.no_alt_analysis_set.fa.gz \
    --reads_normal = ${INPUT_DIR}/${NORMAL} \
    --reads_tumor = ${INPUT_DIR}/${TUMOR} \
    --output_vcf = ${OUTPUT_DIR}/${VCF} \
    --sample_name_tumor = "tumor" \
    --sample_name_normal = "normal" \
    --num_shards = $(nproc)
```

In each case, normal files were a ~30x mix of HG003 and HG004. In the case of short read variant calling (Illumina, Element, Onso) the normal used was NovaSeq (due to my judgment that it wouldn't matter much and because we don't have HG003/4 for Onso). The normal files used are publicly downloadable by these links.

- https://storage.googleapis.com/brain-genomics/awcarroll/giab/mosaic/bams/HG003-HG004.normal.novaseq.grch38.bam
- https://storage.googleapis.com/brain-genomics/awcarroll/giab/mosaic/bams/HG003-HG004.normal.novaseq.grch38.bam.bai
- https://storage.googleapis.com/brain-genomics/awcarroll/giab/mosaic/bams/HG003-HG004.normal.pacbio.grch38.bam
- https://storage.googleapis.com/brain-genomics/awcarroll/giab/mosaic/bams/HG003-HG004.normal.pacbio.grch38.bam.bai

Curation Decisions: Finally, using the KEEP/?/REMOVE column from the mosaic benchmark curation sheet, we declared: If our procedure defined a variant as TP and it was marked as KEEP, our verdict was to KEEP it. If our procedure defined a variant as FP and it was marked as REMOVE, our verdict was to REMOVE it. Otherwise, we perform a manual curation with IGV to complement our verdict in the cases of discordance between our verdict and the GIAB call.

*Genomics division at ITER*

Input Data: For this benchmark, we used Illumina whole genome (WGS) data 2x150bp 300X per individual FASTQ files (HG002, HG003, and HG004 samples) downloaded from the GIAB repository (https://ftp-trace.ncbi.nlm.nih.gov/ReferenceSamples/giab/data_indexes/AshkenazimTrio/sequence.index.AJtrio_Illumina300X_wgs_07292015_updated).

Read Alignment: Each replicate from each run was processed using the following steps from the GATK Best Practices guidelines: BWA to generate alignments in SAM format using hg38 obtained from the GATK bundle.

```
ref = "Homo_sapiens_assembly38.fasta"
bwa mem -K 100000000 -p -v 3 -t 16 -Y ${ref} "HG002.run1.rep1.GRCh38.300x.unmapped.bam" | \
  gatk MergeBamAlignment \
    -ALIGNED/dev/stdin \
    -UNMAPPED "HG002.run1.rep1.GRCh38.300x.unmapped.bam" \
    -O "HG002.run1.rep1.GRCh38.300x.BWA.bam" \
    -R ${ref} \
    -SO "unsorted" \
    --CREATE_INDEX true \
    --ADD_MATE_CIGAR true \
    --CLIP_ADAPTERS false \
    --CLIP_OVERLAPPING_READS true \
    --INCLUDE_SECONDARY_ALIGNMENTS true \
    --MAX_INSERTIONS_OR_DELETIONS -1 \
    --PRIMARY_ALIGNMENT_STRATEGY MostDistant \
    --ATTRIBUTES_TO_RETAIN XS \
    --VALIDATION_STRINGENCY SILENT \
    --EXPECTED_ORIENTATIONS FR \
    --MAX_RECORDS_IN_RAM 2000000 \
    --PROGRAM_RECORD_ID "bwamem" \
    --PROGRAM_GROUP_VERSION "0.7.17-r1188" \
    --PROGRAM_GROUP_COMMAND_LINE "-K 100000000 -p -v 3 -t 16 -Y ${ref}" \
    --PROGRAM_GROUP_NAME "bwamem" \
    --UNMAPPED_READ_STRATEGY COPY_TO_TAG \
    --ALIGNER_PROPER_PAIR_FLAGS true \
    --UNMAP_CONTAMINANT_READS true.
```

Mark duplicates

```
gatk MarkDuplicates \
  -I "HG002.run1.rep1.GRCh38.300x.BWA.bam" \
  -O "HG002.run1.rep1.GRCh38.300x.BWA.deduped.bam" \
  -M "HG002.run1.rep1.GRCh38.300x.BWA.deduped.metrics" \
  --REMOVE_DUPLICATES false \
  --OPTICAL_DUPLICATE_PIXEL_DISTANCE 2500 \
  --VALIDATION_STRINGENCY SILENT \
  --ASSUME_SORT_ORDER queryname \
  --CREATE_MD5_FILE true \
  --CLEAR_DT false.
```

Base quality score recalibration (BQSR)

```
# Analyze patterns of covariation in the sequence dataset for BQSR
gatk BaseRecalibrator \
  -R ${ref} \
  -I "HG002.run1.rep1.GRCh38.300x.BWA.deduped.bam" \
  --use-original-qualities \
  -O "HG002.run1.rep1.GRCh38.300x.BWA.deduped.recal_data.table" \
  --known-sites "dbsnp_146.hg38.vcf" \
  --known-sites "Mills_and_1000G_gold_standard.indels.hg38.vcf"
# Apply the recalibration to your sequence data
gatk ApplyBQSR \
```

```
    -R ${ref} \
    -I "HG002.run1.rep1.GRCh38.300x.BWA.deduped.bam" \
    --use-original-qualities \
    --static-quantized-quals 10 \
    --static-quantized-quals 20 \
    --static-quantized-quals 30 \
    -bqsr "HG002.run1.rep1.GRCh38.300x.BWA.deduped.recal_data.table" \
    --create-output-bam-index \
    --create-output-bam-md5 \
    --add-output-sam-program-record \
    -O "HG002.run1.rep1.GRCh38.300x.BWA.deduped.recal.bam"
```

Replicates per run from the same sample were merged using MergeSamFiles from GATK. Final BAM files per run for each sample were merged using MarkDuplicates from GATK.

```
    gatk MergeSamFiles \
    -I "HG002.run1.rep1.GRCh38.300x.BWA.deduped.recal.bam" \
    -I "HG002.run1.rep2.GRCh38.300x.BWA.deduped.recal.bam" \
    -I "HG002.run1.rep3.GRCh38.300x.BWA.deduped.recal.bam" \
    -O "HG002.run1.merged.GRCh38.300x.BWA.deduped.recal.bam" \
    --ASSUME_SORTED true \
    --SORT_ORDER coordinate \
    --CREATE_INDEX true \
    --REFERENCE_SEQUENCE ${ref} \
    --VALIDATION_STRINGENCY SILENT
    gatk MarkDuplicates \
    -I "HG002.run1.merged.GRCh38.300x.BWA.deduped.recal.bam" \
    -I "HG002.run2.merged.GRCh38.300x.BWA.deduped.recal.bam" \
    -O "HG002.GRCh38.300x.ITER.bam" \
    --CREATE_INDEX true \
    -M "HG002.GRCh38.300x.ITER.metrics" \
    --REMOVE_DUPLICATES false \
    --OPTICAL_DUPLICATE_PIXEL_DISTANCE 2500 \
    --VALIDATION_STRINGENCY SILENT \
    --ASSUME_SORT_ORDER coordinate \
    --CREATE_MD5_FILE true \
    --COMPRESSION_LEVEL 6.
```

Tool Versions: BWA v0.7.15-r1188 (GitHub: https://github.com/lh3/bwa), GATK4 v4.2.0.0 (https://gatk.broadinstitute.org/hc/en-us)

Tumor-only calling Mutect2 with default databases for germline resources and a panel of normals. Individual callers used: MuTect2 (4.4.0.0), SomaticSniper (1.0.5.0), Strelka2 (2.9.5), MuSE (v1.0rc), and VarDict (v1.5.1).

```
    gatk Mutect2 \
    -R ${ref} \
    -I HG002.GRCh38.300x.ITER.bam \
    -tumor HG002 \
    --germline-resource af-only-gnomad.hg38.vcf.gz \
    --panel-of-normals 1000g_pon.hg38.vcf.gz \
    -O HG002.ITER.unfiltered.vcf.gz.
```

Curation Decisions: Finally, using the KEEP/?/REMOVE column from the mosaic benchmark curation sheet, we declared: If our procedure defined a variant as TP and it was marked as KEEP, our verdict was to KEEP it. If our procedure defined a variant as FP and it was marked as REMOVE, our verdict was to REMOVE it. Otherwise, we perform a manual curation with IGV to complement our verdict in the cases of discordance between our verdict and the GIAB call.

*Boutros lab, UCLA.* Data Validation: All pipelines implemented in this project utilize PipeVal (v4.0.0-rc.2) to validate input and output files.[48]

Alignment: Sequence reads were aligned to the GRCh38 reference genome, including decoy contigs (Broad Institute, 2016-07-21), using BWA-MEM2 (v2.2.1).[58] The alignment process was conducted with default settings (i.e., without alternate-contig awareness). Duplicate reads were marked using Picard's MarkDuplicates (v3.0.0) (Picard Toolkit 2019). The Genome Analysis Toolkit (GATK) was used to perform Indel Realignment (v3.7.0) and Base Quality Score Recalibration (BQSR) (v4.2.4.1).[49] After BQSR, the AJ parental

BAM files (HG003 and HG004) were merged using Picard's MergeSamFiles (v3.0.0). The merged BAM underwent header modification using SAMtools reheader (v1.15.1) to replace parental sample IDs (HG003, HG004) with an ID derived from the AJ son and designated as "HG002-N".[59]

Variant Calling: The BQSR BAM of the AJ son HG002 was treated as the tumor sample while the merged parental BAM, HG002-N was treated as the normal sample. Somatic variant calling was performed using the call-sSNV (v7.0.0) pipeline with the tumor/normal BQSR BAMs.[48] The pipeline has two main steps: (1) calling somatic variants using four different somatic variant callers: Mutect2 (v4.4.0.0), SomaticSniper (v1.0.5.0), Strelka2 (v2.9.10) and MuSE (v2.0.4)[33,45,46,49] and (2) intersecting the resulting variant calls using BCFtools (v1.17)[59] to produce consensus variants detected by at least two or more callers. Both consensus and individual caller variants were considered for the HG002 somatic mosaic benchmark evaluation. All alignment and varying calling steps were implemented in Nextflow-based pipelines (Patel et al. in preparation).

Input Data: For this benchmark evaluation, we used the GIAB AJ trio (HG002/HG003/HG004) Illumina PCR-free WGS (2x150bp, 300X) FASTQs.

Read Alignment: Each replicate from each run was processed using the following steps:

Alignments (BWA-MEM2 v2.2.1 & SAMtools v1.12)

```
bwa-mem2 mem -R "@RG\tID:${read_group_id}.Seq${lane}\tCN:${sequencing_center}\
tLB:${library_id}\tPL: ${platform_technology}\tPU:${platform_unit}\tSM:$
{sample}" reference-GRCh38.fa R1.fastq R2.fastq | samtools view -S -b >
lane.bam.
```

Sort alignments from each lane BAM (SAMtools v1.15.1)

```
samtools sort -O bam -o sorted-lane.bam lane.bam.
```

Merge lane BAMs for each sample (SAMtools v1.15.1):

```
samtools merge --write-index -o sample.bai sorted-lane{1.n}.bam.
```

Mark duplicates (Picard v3.0.0):

```
java -jar picard.jar MarkDuplicates --VALIDATION_STRINGENCY LENIENT -INPUT
    sample.bam -OUTPUT sample_markdup.bam --METRICS_FILE markdup_bam.metrics
    --ASSUME_SORT_ORDER coordinate --PROGRAM_RECORD_ID MarkDuplicates --
CREATE_INDEX true.
```

Indel realignment (GATK v3.7.0):

```
java -jar GenomeAnalysisTK.jar --analysis_type RealignerTargetCreator
    sample_markdup.bam --reference_sequence Homo_sapiens_assembly38.fasta --
    knownAlleles Mills_and_1000G_gold_standard.indels.hg38.vcf.gz --
    knownAlleles Homo_sapiens_assembly38.known_indels.vcf.gz --
    allow_potentially_misencoded_quality_scores --targetIntervals
    sample_chr{n}.intervals --out sample_indelrealigned-chr{n}.bam
    --intervals chr{n}-contig.interval_list
```

Base quality score recalibration (GATK v4.2.4.1):

```
gatk BaseRecalibrator sample-indel-realign{1.n}.bam --reference reference-
    GRCh38.fa --verbosity INFO --known-sites
    Mills_and_1000G_gold_standard.indels.hg38.vcf.gz --known-sites
    Homo_sapiens_assembly38.known_indels.vcf.gz --known-sites
    bundle_v0_dbsnp138.vcf.gz --output sample_recalibration_table.grp --
    read-filter SampleReadFilter --sample sample-id
gatk ApplyBQSR -input sample-indel-realign{n}.bam --bqsr-recal-file
    sample_recalibration_table.grp -reference reference-GRCh38.fa --read-
    filter SampleReadFilter --output stdout --sample sample 2>.command.err
    | samtools view -h | awk '(/ ^@RG/&&/SM:sample/) || !/^@RG/' |
    samtools view -b -o sample-bqsr-chr{n}.bam.
```

Tool Versions: BCFtools v1.17, BWA-MEM2 v2.2.1, call-sSNV v7.0.0, GATK v3.7.0, GATK v4.2.4.1,Manta v1.6.0, MuSE v2.0.4, PipeVal v4.0.0-rc.2, Picard v3.0.0, SAMtools v1.12, SAMtools v1.15.1. SomaticSniper v1.0.5.0, and Strelka2 v2.9.10.

Normal Sample Creation (HG002-N)

Merge HG003 and HG004 BAMs (Picard v3.0.0):

```
java -jar picard.jar MergeSamFiles I=HG003.bam I=HG004.bam O=HG003-HG004-
    merged.bam CREATE_INDEX = true.
```

Reheader merged HG003-HG004 BAM to HG002-N (SAMtools v1.15.1):

```
samtools reheader HG002-N.header HG003-HG004-merged.bam > HG002-N.bam.
```

Variant Calling: From the call-sSNV v7.0.0 pipeline.

MuSE v2.0.4.

```
MuSE call -f reference-GRCh38.fa -O MuSE-HG002-T HG002-T.bam HG002-N.bam.
    MuSE sump -I MuSE-HG002-T.txt -G -O MuSE-HG002-T-raw.vcf -D dbsnp.vcf.gz.
```

Mutect2 v4.4.0.0.

GATK Mutect2 Workflow (https://gatk.broadinstitute.org/hc/en-us/articles/360035531132–How-to-Call-somatic-mutations-using-GATK4-Mutect2)

SomaticSniper v1.0.5.0 (downstream filtering not included below)

```
bam-somaticsniper -q 1 '# map_qual 1 is recommended' -Q 15 '# somatic_qual
    default to 15' -T 0.85 '# theta default to 0.85' -N 2 '# haplotypes
    default to 2' -r 0.001 '# prior_haplotypes default to 0.001' -F vcf '#
    output_format here is vcf' '# The next 2 lines are included because in
    the original script 'use_prior_prob' was turned on' -J -s 0.01 -f
    reference-GRCh38.fa HG002-T.bam HG002-N.bam SomaticSniper-HG002-T.vcf.
```

Strelka2 v2.9.10 + Manta v1.6.0

```
configureStrelkaSomaticWorkflow.py –normalBam HG002-
    N.bam -tumorBam HG002-T.bam --referenceFasta reference-GRCh38.fa -indelCandidates Manta-Indel-
    candidates.vcf --runDir StrelkaSomaticWorkflow
```

BCFtools v1.17 to create consensus somatic calls

```
bcftools isec --nfiles +2 --output-type z --prefix isec-2-or-more ${vcf-list}
bcftools --output-type v --output BCFtools-HG002-T_SNV-concat.vcf --
    allow-overlaps --rm-dups all ${vcf-list-from-above-step}
```

Curation Decisions: Finally, using the KEEP/?/REMOVE column from the mosaic benchmark curation sheet, we declared: If our procedure defined a variant as TP and it was marked as KEEP, our verdict was to KEEP it. If our procedure defined a variant as FP and it was marked as REMOVE, our verdict was to REMOVE it. Otherwise, we perform a manual curation with IGV to complement our verdict in the cases of discordance between our verdict and the GIAB call.

