## [Document S2. Transparent peer review records for Daniels et al. · Cell Genomics]

Characterization of subclonal variants in HG002 Genome In A Bottle reference material as a resource for benchmarking variant callers.

Camille A. Daniels, Adetola Abdulkadir, Megan H. Cleveland, Jennifer H. McDaniel, David Jáspez, Luis Alberto Rubio-Rodríguez, Adrián Muñoz-Barrera, José Miguel Lorenzo-Salazar, Carlos Flores, Byunggil Yoo, Sayed Mohammad Ebrahim Sahraeian, Yina Wang, Massimiliano Rossi, Arun Visvanath, Lisa Murray, Wei-Ting Chen, Severine Catreux, James Han, Rami Mehio, Gavin Parnaby, Andrew Carroll, Pi-Chuan Chang, Kishwar Shafin, Daniel Cook, Alexey Kolesnikov, Lucas Brambrink, Mohammed Faizal Eeman Mootor, Yash Patel, Takafumi N. Yamaguchi, Paul C. Boutros, Karolina Sienkiewicz, Jonathan Foon, Christopher E. Mason, Bryan R. Lajoie, Carlos A. Ruiz-Perez, Semyon Kruglyak, Justin M. Zook, and Nathan D. Olson

Summary

Initial submission: Received : Dec 10, 2024

Scientific editor: Sara Rohban

First round of review: Number of reviewers: 2
Revision invited : Jan 15, 2025
Revision received : Apr 18, 2025

Second round of review: Number of reviewers: 2
Revision invited : May 16, 2025
Revision received : Jul 03, 2025

Third round of review: Number of reviewers: 1
Accepted : Nov 21, 2025

Data freely available: YES

Code freely available: YES

This transparent peer review record is not systematically proofread, type-set, or edited. Special characters, formatting, and equations may fail to render properly. Standard procedural text within the editor's letters has been deleted for the sake of brevity, but all official correspondence specific to the manuscript has been preserved.

Referees' reports, first round of review

Reviewer #1:

The manuscript presents a benchmarking dataset for detecting low-frequency mosaic SNVs in the HG002 Genome In A Bottle (GIAB) NIST reference material. Benchmarking datasets are critical for evaluating the accuracy of variant detection tools. While there are existing benchmarking datasets derived from tumor and normal bulk sequencing data, the use of the GIAB sample could be of interest to many people.

However, I have several concerns and suggestions that I believe the authors should address to improve the quality and robustness of their study:

1. Tool Diversity for Mosaic Variant Detection:

The authors detected mosaic variants using only Strelka2. While Strelka2 is a widely used and reliable tool, no single software tool has 100% sensitivity or specificity due to inherent limitations in algorithms and thresholds. Given the availability of multiple tools for mosaic variant detection, I strongly suggest the authors include additional software in their benchmarking pipeline. Incorporating multiple tools would provide a more comprehensive assessment of mosaic variant detection and mitigate the risk of missing true-positive variants that might be overlooked by Strelka2 alone.

2. Exclusion of Mosaic SNVs Based on Long-Read Support:

In the manually curated set of potential mosaic variants, a large fraction of variants were excluded due to the lack of long-read sequencing support. This decision is confusing and warrants further explanation, particularly since these excluded variants were supported by multiple short-read sequencing platforms, including BGI, Illumina, and Element.

The authors focus on short variants (e.g., SNVs) rather than structural variants or mutations in repetitive regions, and short-read sequencing is generally considered to perform better for detecting such mutations due to higher base-level accuracy. It is unclear why long-read support was required for validation in this context. Could there be specific shortcomings associated with the long-read data, such as insufficient coverage, batch effects, or platform-specific biases? These possibilities should be carefully considered and discussed. If the authors' rationale for relying on long-read support stems from a technical or biological consideration, it should be explicitly stated and justified in the manuscript.

Overall, while the benchmarking dataset constructed with GIAB reference material is a valuable resource, addressing the above points would significantly enhance the robustness and comprehensiveness of the study.

Reviewer #2:

This study establishes a benchmark for detecting low-frequency mosaic variants using the HG002 Genome In A Bottle (GIAB) NIST reference material. By employing high-coverage sequencing and the Strelka2 variant caller, the authors identified and validated 85 high-confidence single nucleotide variants (SNVs) with variant allele frequencies (VAF) ranging from 5% to 30% across multiple sequencing platforms. The benchmark is designed to support somatic and mosaic variant detection methods while addressing batch effects in DNA samples. The availability of such a benchmark dataset is undoubtedly valuable to the research community, particularly since it is derived from publicly accessible reference standard materials (GIAB) with donor consent. The authors' efforts in generating this resource are commendable, and the resource merits publication. On the other hand, I have several comments regarding the methodology, presentation, and core contributions of the study that the authors should consider.

Major comments:

1. While the authors assert that they conducted a robust benchmarking process, the work appears more akin to the generation of a robust resource with truth set. The methodology for truth set generation seems thorough, but the manuscript falls short in providing sufficient information on how the truth set was used to evaluate mosaic variant detection methods. Therefore, the contribution of this study should be more focused and properly presented as a resource paper, instead of a benchmark paper.

2. There are notable discrepancies between the goals of the study and the contributions presented in the title and abstract/introduction. Considering the samples and the targeted VAF range (5-30%), the benchmark call sets predominantly represent postzygotic mosaic mutations that arise during early development. While I acknowledge that this VAF range is lower than that of heterozygous germline mutations (~50%), the primary challenge in calling such variants is not the low VAF itself. Instead, the difficulty lies in distinguishing these variants from non-variant positions and heterozygous germline mutations (e.g., MosaicHunter, MosaicForecast, and DeepMosaic). Typical low-VAF calling problems, as addressed in current mosaic variant calling algorithms focus on variants with a VAF of 0.5-2%.

The authors should reconsider the title and revise the introduction to better align with the scope and focus of the study, thereby addressing this discrepancy.

3. The authors compare this study to previous studies including cell line mixing and spike-ins. While this study provides robust, multi-platform validated true variant sets, the small number of a truth set can be limitation. The pros and cons should be described without bias.

4. It is hard to understand how the inclusion of "medically relevant genes" and the batch effect are relevant to the main manuscript. Please describe the purpose of this analysis and why they are important in this study. Alternatively, the authors can more focus on the main theme and provide more detailed workflow and methodologies to support the robustness of the call set.

5. In validating mosaic variant candidates, did the authors select only those supported by all orthogonal sequencing technologies? Given platform-specific errors, such as erroneous indel generation in homopolymer regions by PacBio, how often and why did a platform fail to identify certain variants?

6. A summary table or figure illustrating the methods used for testing would enhance reader comprehension. While the methods section describes six variant calling approaches, the manuscript lacks sufficient discussion on the results of each method and the differences or similarities among their variant call sets.

7. The authors report that 87% of the truth set was identified in data from six groups. What accounts for the remaining 13%, and why were those variants missed? A comparison of call sets across short- and long-read platforms from the six groups would provide valuable insights.

8. In Results, the authors mentioned that "we excluded genomic regions with tandem repeats and homopolymers, regions containing variants that could not be confidently determined to be $>5\%$ or $<2\%$ VAF". But in "Benchmark variant and region characteristics" section, the authors wrote "The benchmark set included variants in challenging genomic regions with two of the 85 variants in homopolymers and two in low mappability regions". Please explain how this can happen.

Minor comments:

1. Please provide line numbers in the manuscript.
 2. Figures could be refined to improve their readability and clarity.
-

Authors' response to the first round of review

We thank the reviewers for their thoughtful feedback and constructive suggestions. We have revised the manuscript as indicated, with all new and modified text shown in blue, and have likewise provided our itemized responses below in blue for clarity. In particular, we have added an expanded analysis of variant allele fraction (VAF) differences between different DNA sources, which further supports our recommendation to use data generated from the NIST RM DNA for the most accurate subclonal benchmarking results. We appreciate the reviewers' help in strengthening our manuscript and addressing important points regarding batch effects and study design.

Reviewers' Comments and Responses:

Reviewer #1: The manuscript presents a benchmarking dataset for detecting low-frequency mosaic SNVs in the HG002 Genome In A Bottle (GIAB) NIST reference material. Benchmarking datasets are critical for evaluating the accuracy of variant detection tools. While there are existing benchmarking datasets derived from tumor and normal bulk sequencing data, the use of the GIAB sample could be of interest to many people.

However, I have several concerns and suggestions that I believe the authors should address to improve the quality and robustness of their study:

1. Tool Diversity for Mosaic Variant Detection:

The authors detected mosaic variants using only Strelka2. While Strelka2 is a widely used and reliable tool, no single software tool has 100% sensitivity or specificity due to inherent limitations in algorithms and thresholds. Given the availability of multiple tools for mosaic variant detection, I strongly suggest the authors include additional software in their benchmarking pipeline. Incorporating multiple tools would provide a more comprehensive assessment of mosaic variant detection and mitigate the risk of missing true-positive variants that might be overlooked by Strelka2 alone.

The Strelka2 tumor-normal callset was used to generate the list of potential mosaic variants and we evaluated the highly sensitive list of 425,679 potential mosaic variants, including filtered variants, in the benchmark regions. Our mixture study found that strelka2 had 99.4% recall for SNVs at 5% VAF when including filtered variants. To clarify this we have added this sentence to the beginning of the results (page 6 lines 186-187) "We kept both PASS and filtered variants from Strelka2 based on the results of the in silico mixture experiments demonstrating 99.4 % recall for SNVs at 5% VAF

(Supplemental Figure 1)." Furthermore, for the external validation of the mosaic benchmark, we used a variety of callers from different technologies to identify mosaic variants that might be

missed by strelka2. Eight external groups produced somatic callsets across a number of variant calling algorithms in tumor-only or tumor-normal modes (Supplementary Table 5) using the same high coverage (300X) GIAB data as input, which support that the mosaic benchmark can reliably identify false positives and meets reliable identification of errors (RIDE) principle.

2. Exclusion of Mosaic SNVs Based on Long-Read Support:

In the manually curated set of potential mosaic variants, a large fraction of variants were excluded due to the lack of long-read sequencing support. This decision is confusing and warrants further explanation, particularly since these excluded variants were supported by multiple short-read sequencing platforms, including BGI, Illumina, and Element.

The authors focus on short variants (e.g., SNVs) rather than structural variants or mutations in repetitive regions, and short-read sequencing is generally considered to perform better for detecting such mutations due to higher base-level accuracy. It is unclear why long-read support was required for validation in this context. Could there be specific shortcomings associated with the long-read data, such as insufficient coverage, batch effects, or platform-specific biases? These possibilities should be carefully considered and discussed. If the authors' rationale for relying on long-read support stems from a technical or biological consideration, it should be explicitly stated and justified in the manuscript.

We clarified our rationale for excluding potential mosaic variants from the HG002 mosaic benchmark (page 6 lines 193 - 208). We clarified that “More than 98% of the candidates were removed due to very low VAF (less than 3% with 99% confidence).” We added this text to mitigate concerns about coverage and batch effects: “We only used technologies with at least 100x coverage to ensure sufficient power to have reads supporting variants >5% VAF, and all data were from the NIST RM DNA to avoid batch effects.” When performing curation, we took into consideration platform-specific biases, such as indel error rate for long reads and mapping errors for short reads. We added this text to clarify this: “For variants excluded due to lack of long-read support, most were detected in one or both parents (HG003 and/or HG004) and were in segmental duplication regions associated with mapping errors and copy number variation.”

Overall, while the benchmarking dataset constructed with GIAB reference material is a valuable resource, addressing the above points would significantly enhance the robustness and comprehensiveness of the study.

Reviewer #2: This study establishes a benchmark for detecting low-frequency mosaic variants using the HG002 Genome In A Bottle (GIAB) NIST reference material. By employing high-coverage sequencing and the Strelka2 variant caller, the authors identified and validated 85 high-confidence single nucleotide variants (SNVs) with variant allele frequencies (VAF) ranging from 5% to 30% across multiple sequencing platforms. The benchmark is designed to support somatic and mosaic variant detection methods while addressing batch effects in DNA samples. The availability of such a benchmark dataset is undoubtedly valuable to the research community, particularly since it is derived from publicly accessible reference standard materials (GIAB) with donor consent. The authors' efforts in generating this resource are commendable, and the resource merits publication. On the other hand, I have several comments regarding the methodology, presentation, and core contributions of the study that the authors should consider.

Major comments:

1. While the authors assert that they conducted a robust benchmarking process, the work appears more akin to the generation of a robust resource with truth set. The methodology for truth set generation seems thorough, but the manuscript falls short in providing sufficient information on how the truth set was used to evaluate mosaic variant detection methods. Therefore, the contribution of this study should be more focused and properly presented as a resource paper, instead of a benchmark paper.

The reviewer is correct that the work we describe in this paper is for the development of a mosaic variant benchmark set (aka "truth set") that is a resource for evaluating and validating low-frequency and somatic variant callers.

The study is not meant as an evaluation of mosaic variant detection methods. The comparison of the benchmark set to mosaic variant calling methods is presented as a validation of the benchmark set ensuring that it follows the RIDE principle (reliably able to identify errors.) Because sequencing and analysis methods are constantly improving, benchmarking of particular methods tends to be outdated quickly. Instead, the Genome In A Bottle Consortium develops robust benchmarks as an enduring resource that anyone can use to evaluate methods now and in the future, which is the goal of this manuscript. As a side note, GIAB uses the term "benchmark set" to describe our reference material characterizations rather than "truth set" or "gold standard" because we are able to confirm the benchmark is fit for the purpose of benchmarking by ensuring it follows the RIDE principles.

To clarify this point in the manuscript we revised the title "Characterization of subclonal variants in HG002 Genome In A Bottle reference material as a resource for benchmarking variant callers." and modified abstract and results text.

Modified abstract (page 2: lines 52 - 56):

“To address the need for benchmarking subclonal variants in normal cell populations, we developed a benchmark set containing mosaic variants in the Genome in a Bottle Consortium (GIAB) HG002 reference material DNA from a large batch of a normal lymphoblastoid cell line *for evaluating lower-frequency variant callsets.*”

Results: Changed the subheading “External Validation” to “Mosaic Benchmark Set External Validation”

Added the following text to the start of the Mosaic Benchmark Set External Validation section (page 11 lines 272-275) “To validate that the mosaic benchmark set can be used to reliably identify errors (RIDE principle), we compared the draft benchmark set to variant callsets submitted by external evaluators, and manually curated the discrepancies ensuring the discrepancies are errors in the comparison callset and not the benchmark.”

2. There are notable discrepancies between the goals of the study and the contributions presented in the title and abstract/introduction. Considering the samples and the targeted VAF range (5-30%), the benchmark call sets predominantly represent postzygotic mosaic mutations that arise during early development. While I acknowledge that this VAF range is lower than that of heterozygous germline mutations (~50%), the primary challenge in calling such variants is not the low VAF itself. Instead, the difficulty lies in distinguishing these variants from non-variant positions and heterozygous germline mutations (e.g., MosaicHunter, MosaicForecast, and DeepMosaic). Typical low-VAF calling problems, as addressed in current mosaic variant calling algorithms focus on variants with a VAF of 0.5-2%.

The authors should reconsider the title and revise the introduction to better align with the scope and focus of the study, thereby addressing this discrepancy.

Modified the title to "Characterization of subclonal variants in HG002 Genome In A Bottle reference material as a resource for benchmarking variant callers." and amended Line 157 to specify 5-30% targeted VAF range for this study.

3. The authors compare this study to previous studies including cell line mixing and spike-ins. While this study provides robust, multi-platform validated true variant sets, the small number of a truth set can be a limitation. The pros and cons should be described without bias.

Indeed, we view this benchmark as complementary to, rather than competing with, previous studies like cell line mixing and spike-ins. In fact, a substantial motivation for developing this benchmark was the use of HG002 as a background in mixing studies and spike-ins, where the lack of characterization of mosaic variants in HG002 made it challenging to distinguish between false positives and true mosaic variants in HG002. We have clarified this and the limitations of the benchmark in the Discussion (page 12 lines 340 - 348) : “A limitation of this benchmark is the relatively small number of true mosaic variants compared to other benchmarking approaches such as sample mixtures, tumor cell lines, and spike-ins. In addition, these variants may not precisely resemble biological mosaic and somatic variants, and are at higher VAF than some biological variants. However, characterizing these variants and excluding potential mosaic variants above 2 % VAF provides a more comprehensive negative control for identifying false positives, and improves the utility of the GIAB germline benchmark for HG002. In addition, it is complementary to approaches such as sample mixtures, spike-ins, and engineering variants by providing a well-characterized background DNA sample that is commonly used in these studies.”

4. It is hard to understand how the inclusion of "medically relevant genes" and the batch effect are relevant to the main manuscript. Please describe the purpose of this analysis and why they are important in this study. Alternatively, the authors can more focus on the main theme and provide more detailed workflow and methodologies to support the robustness of the call set.

The presence or absence of medically relevant variants medically relevant genes analysis is of interest to users in the clinical setting when validating diagnostic assays.

Removed the heading for the batch effects section and incorporated text into the previous “benchmark characterization” section. The following text was added to the beginning and end of the paragraph to provide context for the importance of this analysis to users of this benchmark, as this is a question GIAB frequently is asked (page 10 lines 247 - 248, 256-258). “The mosaic benchmark set provides a characterization and VAFs for variants in the NIST RM 8391 DNA, but these can differ in cells available from other batches of HG002. For example, we ... For accurate benchmarking, users will want to use variants called using sequencing data generated from the NIST RM, as other batches may have different VAF profiles or sets of low-frequency variants.” Additionally, we revised the statistical analysis of benchmark variants VAFs between sequencing data generated from NIST RM DNA and non-RM DNA. First we test for global differences in variant VAF between the different DNA sources using a generalized linear mixed-effects model, modeling the number of reads supporting the alternate and reference allele per variant using a binomial distribution. DNA source and sequencing method were modeled as fixed effects and variant as

random effect. Individual variants with statistically significant differences in VAF between DNA source using separate binomial generalized linear models for each variant. Figure 4 was revised to highlight variants with significantly different VAFs.

5. In validating mosaic variant candidates, did the authors select only those supported by all orthogonal sequencing technologies? Given platform-specific errors, such as erroneous indel generation in homopolymer regions by PacBio, how often and why did a platform fail to identify certain variants?

After filtering using heuristics and manual curation in IGV, variants included in the mosaic benchmark were detected by all ortho technologies. A labelling discrepancy in Supplemental Table S3 Column O was amended to reflect this result. In general, while we expect platform-specific systematic sequencing errors to lead to false positives, they typically do not result in lack of evidence for a variant because all datasets have at least 100x coverage, sufficient to see variants at 5% VAF unless there are coverage biases.

6. A summary table or figure illustrating the methods used for testing would enhance reader comprehension. While the methods section describes six variant calling approaches, the manuscript lacks sufficient discussion on the results of each method and the differences or similarities among their variant call sets.

This manuscript focuses on the development, characterization, and validation of the mosaic benchmark set and not a somatic variant caller comparison (see response to major comment 1) therefore we do not provide results on the differences and similarity among the different methods.

7. The authors report that >87% of the truth set was identified in data from six groups. What accounts for the remaining 13%, and why were those variants missed? A comparison of call sets across short- and long-read platforms from the six groups would provide valuable insights.

We revised the text to include the following text (pages 11-12, lines 310 - 314) "A total of 16 variants in the benchmark set were not identified by all external callsets (using > 100X coverage data as input.) All 16 variants had VAFs without our level of detection (5-30% VAF) with 11 of the 16 were missed by one shortread callset. The 16 variants were manually curated (internally and by multiple external groups) and confirmed to be true mosaic variants. " The focus of our study is evaluation of the benchmark set rather than a comparison of variant callers so we did not compare the callsets.

8. In Results, the authors mentioned that "we excluded genomic regions with tandem repeats and homopolymers, regions containing variants that could not be confidently determined to be >5% or <2% VAF". But in "Benchmark variant and region characteristics" section, the authors wrote "The benchmark set included variants in challenging genomic regions with two of the 85 variants in homopolymers and two in low mappability regions". Please explain how this can happen.

Thank you for highlighting this apparent contradiction. Investigating the two benchmark variants "in homopolymers", it turns out that they are 5bp away from a homopolymer, at the boundary of the bed files we used, which include 5bp flanking regions around homopolymers to include potential artifacts around homopolymers. Because some technologies may have sequencing errors due to being near the homopolymer, we changed this text (page 8 lines 225 - 226) to

"The benchmark set included variants in challenging genomic regions with two of the 85 variants *near* homopolymers". Additionally, we clarified the bed files used to define the benchmark regions (page 18, lines 581-582 and 585-586.)

Minor comments:

1. Please provide line numbers in the manuscript. Line numbers added
2. Figures could be refined to improve their readability and clarity.

Figure revisions:

- Figure 2 - X and Y axis labels were simplified, the point annotations were removed, and the figure legend was revised.
 - Figure 3 - Removed panel B in figure 3 as relevant numbers are stated in text. For panel A we removed colors changed to vertical layout for easier comparison of VAF between variants included and excluded from the mosaic benchmark and increased font size for improved readability. Also, revised figure legend for clarity.
 - Figure 4 - Removed sequencing method coloring and modified line and point color to highlight variants in the mosaic benchmark with VAFs statistically different between sequencing datasets generated using RM DNA and non-RM DNA.
-

Referees' reports, second round of review

Reviewer #1: My concerns have been appropriately addressed.

Reviewer #2: The authors resolved most of my previous concerns, and the manuscript has been significantly improved. I now have some minor points that should be resolved before publication.

Comments:

1. Summarizing the results of the Mosaic Benchmark Set External Validation section in a figure or table would help improve understanding.
2. While the call set comparison itself may not be included in this part, the methods used (e.g., DRAGEN, DeepSomatic, Sentieon, etc.) are only mentioned in the supplementary materials. It would be helpful to briefly mention them in the main text as well.
3. The current truth set includes only SNVs, with all indels removed. The reason for this should be clarified.
4. The abstract repeatedly mentions "mosaic variants," but since this paper specifically deals with SNVs, clearly stating this would help avoid confusion.

Authors' response to the second round of review

Thank you very much for your thoughtful feedback and helpful suggestions. We appreciate the opportunity to improve our manuscript and have carefully addressed the points raised.

Reviewer #2: The authors resolved most of my previous concerns, and the manuscript has been significantly improved. I now have some minor points that should be resolved before publication.

1. Summarizing the results of the Mosaic Benchmark Set External Validation section in a figure or table would help improve understanding.

- Thank you for this excellent suggestion. We have added both a new Figure 5 and Table 1 presenting the external validation results. These additions show how the external callsets validate that the mosaic benchmark set meets the RIDE principles for reliably identifying both false positives and false negatives.

2. While the call set comparison itself may not be included in this part, the methods used (e.g., DRAGEN, DeepSomatic, Sentieon, etc.) are only mentioned in the supplementary materials. It would be helpful to briefly mention them in the main text as

- We added the following summary text to the methods section to briefly describe the external validation data and variant callers:

“Briefly, external collaborators used publicly available data generated using the HG002, HG003, and HG004 NIST reference material DNA: Illumina, Pacbio HiFi, Element, Ultima, or Onso (non-RM DNA). HG002 was used as the tumor sample and groups who performed tumor-normal sequencing used either HG003, combined HG003 and HG004, or HG004 as the normal sample. The 8 groups used different variant callers including Illumina’s DRAGEN pipeline, the deep-learning model DeepSomatic,³⁴ Sentieon TNScope,³⁵ and widely used heuristic or probabilistic tools such as Mutect2,³⁶ Strelka2,³⁷ VarDict,³⁸ and Lancet.³⁹ Detailed descriptions of the methods used by the eight groups are available in the Supplemental Methods.”

3. The current truth set includes only SNVs, with all indels removed. The reason for this should be clarified.

To clarify, we added this sentence to the manuscript (lines 226-227):

“All of the candidate indels identified during the benchmark set generation process were excluded by our heuristics and therefore the benchmark set only includes SNVs.”

4. The abstract repeatedly mentions "mosaic variants," but since this paper specifically deals with SNVs, clearly stating this would help avoid confusion.

We replaced generic references to “variants” with “single nucleotide variants (SNVs)” in the abstract to clarify the scope of the benchmark set.

We thank the reviewer again for these constructive comments which improved the clarity and completeness of our manuscript.

Referees’ reports, third round of review

Reviewer #2: The authors successfully addressed all of my previous concerns.